# *Warped Diffusion:* Solving Video Inverse Problems with Image Diffusion Models

**Giannis Daras**[*]
UT Austin

**Weili Nie**
NVIDIA

**Karsten Kreis**
NVIDIA

**Alexandros G. Dimakis**
UT Austin

**Morteza Mardani**
NVIDIA

**Nikola B. Kovachki**
NVIDIA

**Arash Vahdat**
NVIDIA

## Abstract

Using image models naively for solving inverse video problems often suffers from flickering, texture-sticking, and temporal inconsistency in generated videos. To tackle these problems, in this paper, we view frames as continuous functions in the 2D space, and videos as a sequence of continuous warping transformations between different frames. This perspective allows us to train function space diffusion models only on *images* and utilize them to solve temporally correlated inverse problems. The function space diffusion models need to be equivariant with respect to the underlying spatial transformations. To ensure temporal consistency, we introduce a simple post-hoc test-time guidance towards (self)-equivariant solutions. Our method allows us to deploy state-of-the-art latent diffusion models such as Stable Diffusion XL to solve video inverse problems. We demonstrate the effectiveness of our method for video inpainting and $8\times$ video super-resolution, outperforming existing techniques based on noise transformations. We provide generated video results in the following URL: https://giannisdaras.github.io/warped_diffusion.github.io/.

## 1 Introduction

Diffusion models (DMs) [79, 39, 82] can synthesize photorealistic imagery [73, 64, 66, 5, 60, 26]. They can be conditioned easily, through explicit training or guidance [25, 41], and have also been widely used to solve inverse problems [19, 86, 15, 16, 80, 84, 47, 56], in particular for image processing applications like inpainting and super-resolution [40, 72, 74, 66].

How do these methods extend to video processing and solving inverse problems on videos? Although video DMs are seeing rapid progress [38, 78, 9, 29, 8, 30, 7, 11], general text-to-video synthesis has not yet reached the level of robustness and expressivity comparable to modern image models. Moreover, no state-of-the-art video generative models are publicly available [11], and

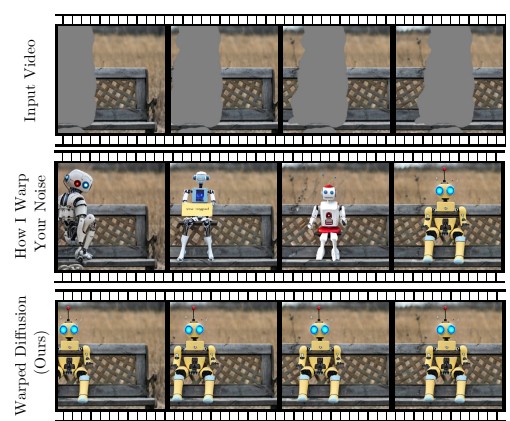

Figure 1: Inpainting results for "a robot sitting on a bench". As the input video shifts smoothly, our output frames stay consistent.

most video DMs are computationally expensive. To circumvent these challenges, a natural research direction is to leverage existing, powerful *image* generative models to solve *video* inverse problems.

---

[*]The work was done during an internship at NVIDIA.

38th Conference on Neural Information Processing Systems (NeurIPS 2024).

Naively applying image DMs to videos in a frame-wise manner violates temporal consistency. Previous works alleviate the problem by fine-tuning on video data or by warping the networks' features, using, for instance, temporal or cross-frame attention layers [88, 54, 13, 51, 61, 90, 92, 34, 33]. However, these methods are usually designed specifically for high-level text-driven editing or stylization and are typically not directly applicable to general inverse problems. Moreover, without training on diverse video data they often cannot maintain high frequency information across frames. For a detailed discussion of the related works, we refer the reader to Section E in the Appendix.

The recent novel work, "How I Warped Your Noise" [14], proposes *noise warping* to achieve temporal consistency in generated videos by changing appropriately the input noise to the diffusion model. Videos can be thought of as image frames subject to spatial transformations. An object may move according to a translation; complex and general transformations can be described by motion vectors on the pixels defined through optical flow [27]. It is these transformations that define how the noise maps need to be warped and transformed. In [14], temporally consistent noise maps are given as input to the DM's denoiser, with the underlying assumption that temporally consistent inputs induce temporally consistent network outputs. In this paper, we argue that this assumption only holds true if the utilized image DM is *equivariant* with respect to the spatial warping transformations. However, as we show in this work, the network is not necessarily equivariant because i) the conditional expectation modeled by the DM may not be equivariant, and, ii) more importantly, a free-form neural network, as used in typical DMs, will not learn a perfectly equivariant function. When the equivariance assumption is violated, the method proposed in [14] achieves poor results. This is typically the case for challenging conditional tasks (see Figure 1) or when modeling complex distributions. Particularly, [14] finds that the proposed method has "limited impact on temporal coherency" when applied to *latent* diffusion models and that "all the noise schemes produce temporally inconsistent results".

We introduce a new framework, dubbed *Warped Diffusion*, for the rigorous application of image DMs to video inverse problems. We employ a continuous function space perspective to DMs [52, 59, 28, 35] that naturally allows noise warping for arbitrarily complex spatial transformations. Our method generalizes the warping scheme of [14] and does not require any auxiliary high-resolution noise maps. To achieve equivariance, we propose *equivariance self-guidance*, a novel sampling mechanism that enforces that the generated frames are consistent under the warping transformation. Our inference time approach elegantly circumvents the need for additional training. This unlocks the use of existing large DMs in a fully equivariant manner without further training, which may be prohibitive for a practitioner.

We extensively validate our method on video inpainting and super-resolution. Super-resolution represents a situation with strong conditioning, while inpainting requires large-scale, temporally coherent synthesis of new content. Warped Diffusion outperforms previous methods quantitatively and qualitatively, and shows reduced flickering and texture sticking artifacts. Due to our equivariance guidance, our method can also be used with *latent* DMs, which is not possible with previous approaches. Virtually all existing state-of-the-art text-to-image generation systems are indeed latent DMs, like Stable Diffusion [66]. Hence, any inverse problem solving method must be readily usable with latent DMs. In fact, all our experiments utilize the state-of-the-art text-to-image latent DM SDXL [60].

**Contributions:** *(a)* We propose Warped Diffusion, a novel framework for applying image DMs to video inverse problems. *(b)* We introduce a principled scheme for noise warping, based on Gaussian processes and a function space DM perspective. *(c)* We identify the equivariance of the DM as a critical requirement for the seamless application of image DMs to video inverse problems and propose an inference-time guidance method to enforce it. *(d)* We comprehensively test Warped Diffusion and achieve state-of-the-art video processing performance when considering the use of image DMs. Critically, Warped Diffusion can be used with any image DMs, including large-scale latent DMs.

## 2 Functional Video Generation

The basis of our approach, summarized in Figure 2, is to structure the generative model so that it is equivariant with respect to spatial deformations and apply these deformations successively to the input noise. Each deformation effectively warps the noise and the equivariance guarantees that each output image will be similarly warped. By using an optical flow from a real video to define a sequence of such deformations, a new video can be generated. To introduce our method, we first conceptualize both images and noise as functions on a domain and the generator as a mapping between two function spaces.

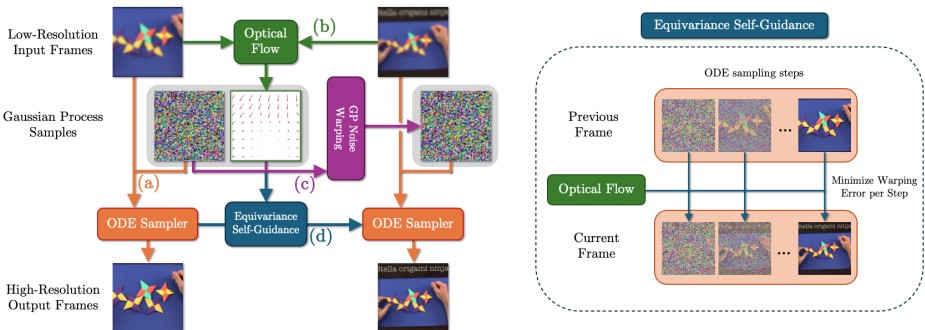

Figure 2: Visualization of Warped Diffusion applied to video super-resolution. (a) We develop a function space diffusion model that super-resolves images given samples from a Gaussian process (GP). To extend the image model to videos, (b) we extract warping transformations between consecutive input frames using optical flow. (c) We use the flow to warp the GP sample from the previous frame. (d) To ensure temporal consistency, we introduce equivariance self-guidance in the ODE sampler.

## 2.1 Functional Generative Modeling and Videos

Each video frame can be seen as a single image, and an image as a discretization of a vector-valued function on a rectangular domain. Consider the domain as the 2-D unit square $D = [0,1]^2$, defining an image as a function $f : D \to \mathbb{R}^3$. For each location $x \in D$, the value $f(x) \in \mathbb{R}^3$ represents an RGB color. We assume images have infinite resolution. To formulate a model that generates such images, we must have a notion of a space containing all possible images. We'll use the separable Hilbert space $H = L^2(D; \mathbb{R}^3)$, with pointwise formulas interpreted almost everywhere with respect to the Lebesgue measure.

We assume that there exists a probability measure $\mu$ on $H$ whose support is the set of photorealistic images and denote by $\eta$ a known reference probability measure on $H$. In our case, $\eta$ will be a Gaussian measure on $H$; for details, see Section 3.1. A *generative model*, or *transport map*, is then a mapping $G : H \to H$ such that the pushforward of $\eta$ under $G$ is $\mu$ which we denote as $G_\sharp \eta = \mu$. In particular, this implies that any random variable $\xi \sim \eta$ will satisfy $G(\xi) \sim \mu$. For diffusion models, $G$ can be defined by the probability flow ODE; see Section 3.2.

Given an image $f_0 \in H$, a *video* with $n+1 \in \mathbb{N}$ frames is the sequence of functions $(f_0, f_1, \ldots, f_n) \in H^{n+1}$, where each subsequent function is obtained, at least partially, from the previous one by a deformation. Specifically, a sequence of bounded, injective maps $(T_j : D \to D_j)_{j=1}^n$ exists such that

$$\underset{\text{Frame Index}}{f_{\,j}}\ (\ \underset{\text{Pixel Location}}{x}\ ) = f_{\,j-1}(\ \underset{\text{Frame of Vision}}{T_{\,j}^{-1}}(\ x\ )), \qquad \forall\, x \in \underset{\text{Deformed Domain}}{D \cap D_j}, \quad j = 1, \ldots, n, \tag{1}$$

where $D_j := T_j(D)$ and we assume that the sets $D \cap D_j$ have positive Lebesgue measure. In video modeling, the sequence $(T_j)_{j=1}^n$ is usually referred to as the *optical flow* as it specifies how each pixel in the previous frame moves to the next frame. While the frames can also be conceptualized as a continuum in time, we work with a discrete set of frames for simplicity. We consider $D$ to always represent our fixed frame of vision and we allow each $T_j$ to move pixels outside of this frame. Therefore (1) determines $f_j$ only on the set $D \cap D_j$ which contains pixels that remain within our field of vision.

## 2.2 Video Generation and Equivariance

Given our notion of a video and a generative model, we now describe how such a model can be used to generate new videos. Suppose we want to create a two-frame video given an initial frame $f_0 \in H$ and a deformation map $T_1 : D \to D_1$. Assume we have a generative model $G : H \to H$ and an initial noise image $\xi_0 \in H$ such that $G(\xi_0) = f_0$. From definition, the new frame of our video is

$f_1 = f_0 \circ T_1^{-1}$ on $D \cap D_1$. If $D \subseteq D_1$, it might seem that our generative model is unnecessary. However, proceeding this way generates blurry and unrealistic videos.

The primary issue is that, in practice, we don't have access to $f_0$ at an infinite resolution but only at a fixed, finite set of grid points $E_k = \{x_1, \ldots, x_k\} \subset D$. To determine $f_1$ on our grid points, we need the values of $f_0$ at the points $T_1^{-1}(E_k) = \{T_1^{-1}(x_1), \ldots, T_1^{-1}(x_k)\}$. It's highly unlikely that $E_k = T_1^{-1}(E_k)$ for any realistic deformation.

Thus, we must interpolate $f_0$ to $T_1^{-1}(E_k)$, which usually leads to blurry results with standard methods. Furthermore, if $D \not\subseteq D_1$, there will be regions where $f_1$ is not determined by $f_0$ and will need to be inpainted on the new visible domain. Therefore, for each frame, we must solve an interpolation and an inpainting problem: tasks for which generative models are well-suited.

Suppose we have access to the noise function $\xi_0$ at infinite resolution, and its domain extends to all of $\mathbb{R}^2$; we discuss both in Section 3.1. We can then define the new frame in our video by applying the generative model to the deformed noise: $f_1 = G(\xi_0 \circ T_1^{-1})$. The deformed noise function $\xi_0 \circ T_1^{-1}$ gets its values from $\xi_0|_D$ for points in $D \cap D_1$ and from the extension of $\xi_0$ to $\mathbb{R}^2$ for all other points where inpainting is needed. To ensure this definition is consistent with (1), $G$ must be equivariant with respect to $T_1^{-1}$. Specifically, for all $\xi \in \text{supp}(\eta) \subseteq H$, we must have

$$G\big(\xi \circ T_1^{-1}\big)(x) = G\big(\xi\big)\big(T_1^{-1}(x)\big), \qquad \forall\, x \in D \cap D_1. \tag{2}$$

Assuming (2), it follows from $G(\xi_0) = f_0$, that $f_1(x) = G(\xi_1|_D)(x) = (f_0 \circ T_1^{-1})(x)$ for all $x \in D \cap D_1$ hence the pair $(f_0, f_1)$ is a valid 2 frame video according to the definition of Section 2.1. To generate a video with any number of frames, we simply iterate on this process with a given sequence of deformation maps. Enforcing (2) can be done directly by the architectural design, through training with various deformation maps, or, through a guidance process; see Section 3.2.

## 2.3 White Noise

It is common practice to train generative models assuming the reference measure $\eta$ is Gaussian white noise. Specifically, a draw $\xi \sim \eta$ on the grid points $E_k = \{x_1, \ldots, x_k\} \subset D$ is realized as $\xi(x_l) = \chi_l$ for an i.i.d. sequence $\chi_l \sim \mathcal{N}(0, 1)$ for $l = 1, \ldots, k$. However, this approach is incompatible with our goal of having the generative model perform interpolation. For most deformations $T$ encountered in practice, none of the points in $T^{-1}(E_k)$ will match those in $E_k$. Consequently, each new evaluation $\xi\big(T^{-1}(x_l)\big)$ will be independent of the sequence $\{\chi_l\}_{l=1}^k$, making $\xi\big(T^{-1}(E_k)\big)$ appear as a new noise realization unrelated to $\xi(E_k)$. This incompatibility arises because white noise processes are distributions, not regular functions, meaning realizations are almost surely not members of $H$ [18]. [14] proposes a stochastic interpolation method to address this issue (see Appendix C for details and comparison). We generalize this idea and propose using generic Gaussian processes on $H$.

## 3 Method: Warped Diffusion

In Section 1, we formulated the problem of video generation as the computation of a series of functions warped by an optical flow and proposed the use of a generative model for inpainting and interpolating the warped functions. The main challenges which remain are defining a functional noise process which can be evaluated continuously and a generative model which is equivariant with respect to warping. We propose to use Gaussian processes for our functional noise and a guidance procedure within the sampling step of a diffusion model to overcome these challenges.

### 3.1 Gaussian Processes (GPs)

A Gaussian Process (GP) $\eta$ is a probability measure on $H$ completely specified by its mean element and covariance operator. For a mathematical introduction, see Appendix B. We identify Gaussian processes with positive-definite kernel functions $\kappa : \mathbb{R}^2 \times \mathbb{R}^2 \to \mathbb{R}$. Recall that $E_k = \{x_1, \ldots, x_k\}$ denotes the grid points where we know the values of an image $f \in H$. To realize a random function $\xi \sim \eta$ on these points, we sample the finite-dimensional multivariate Gaussian $N(0, Q)$, where $Q \in \mathbb{R}^{k \times k}$ is the kernel matrix $Q_{ij} = \kappa(x_i, x_j)$ for $i, j = 1, \ldots, k$.

Once sampled, given the fixed values $\xi(E_k)$, $\xi$ can be evaluated at any new point $x^* \in D$ by computing the conditional distribution $\xi(x^*) \mid \xi(E_k)$ [65]. This approach allows us to realize

random functional samples at infinite resolution through conditioning, thus resolving the interpolation problem. Furthermore, by ensuring the kernel $\kappa$ is positive definite on a domain larger than $D$, we can consistently sample $\xi$ outside of $D$, addressing the inpainting problem described in Section 2.2.

For high-resolution images when $k$ is large, working with the matrix $Q$ can be computationally expensive. Instead, we propose using **R**andom **F**ourier **F**eatures (RFF) to sample $\eta$, which amounts to a finite-dimensional projection of the function $\xi \sim \eta$ that converges in the limit of infinite features [63, 87]. We can approximate samples from a GP with a squared exponential kernel with length-scale parameter $\epsilon > 0$ by $\xi(x) = \sqrt{\frac{2}{J}} \sum_{j=1}^{J} w_j \cos\left(\langle z_j, x \rangle + b_j\right)$ for i.i.d. sequences $w_j \sim N(0, 1)$, $z_j \sim N(0, \epsilon^{-2} I_2)$, $b_j \sim U(0, 2\pi)$ where $J \in \mathbb{N}$ is the number of features. RFF allows us access to $\xi$ at infinite resolution on the entirety of the plane while also allowing for efficient computation.

## 3.2 Function Space Diffusion Models and Equivariance Self-Guidance

We will now focus on the generative model that needs to be equivariant to the noise transformations. Specifically, in this section, i) we introduce function space diffusion models, ii) we prove that if every prediction of the diffusion model is equivariant then the whole diffusion model sampling chain is equivariant to the underlying spatial transformations, and, iii) we describe *equivariance self-guidance*, our sampling technique for enforcing the equivariance assumption.

For ease of notation, we will present everything for the case of unconditional video generation. However, our method seamlessly incorporates any addition conditioning information that may be available. If $c_0, \ldots, c_n \in \mathbb{R}^c$ is a sequence of known conditioning vectors then these can simply be passed into a conditional score model at the appropriate frame without any other change to our method; see Algorithm 1. Conditioning vectors could be, for example, low resolutions versions of a video or an original video with regions masked. In Section 4, we focus on such conditional tasks.

**Function Space Diffusion Models.** Typically, diffusion models are trained with white noise. As explained in Section 2.3, a principled continuous evaluation of the noise requires a functional process. We briefly describe diffusion models in the context of sampling using the Gaussian processes of Section 3.1. We show in Section 4.1 (Table 1) that a model trained with white noise can be fine-tuned to GP noise without any loss in performance.

While it is possible to formulate diffusion models on the infinite-dimensional space $H$ e.g. [52], we will proceed in the finite-dimensional case for ease of exposition. In particular, we will define the forward and backward process as a flow on a vector $u \in \mathbb{R}^k$, thinking of the entries as the values of a scalar function evaluated on the grid $E_k$ and recall that $Q$ is the kernel matrix on $E_k$.

We consider forward processes of the form,

$$\mathrm{d}u_t = \left(2\sigma(t)\dot{\sigma}(t)Q\right)^{1/2}\mathrm{d}W_t, \quad u(0) = u_0 \sim \mu \tag{3}$$

where $W_t$ is a standard Wiener process on $\mathbb{R}^k$ and $\sigma$ is a scalar-valued, once differentiable function. This process results in conditional distributions $p(u_t|u_0) = N(u_0, \sigma^2(t)Q)$, see [46]. Let $p(u_t, t)$ denote the density of $u_t$ induced by (3). Then the following backward in time ODE,

$$\frac{\mathrm{d}u_t}{\mathrm{d}t} = -\sigma(t)\dot{\sigma}(t)Q\nabla_u \log p(u_t, t) \tag{4}$$

started at $u(\tau)$ distributed according to (3) has the same marginal distributions $p(u_t, t)$ as (3) on the interval $[0, \tau]$; see [46]. Approximating $N(u_0, \sigma^2(\tau)Q)$ by $N(0, \sigma^2(\tau)Q)$, we may then define the generative model $G$ by the mapping $u(\tau) \mapsto u(0)$ with reference measure $\eta = N(0, \sigma^2(\tau)Q)$.

Solving (4) requires knowledge of the score $\nabla_u \log p(u_t, t)$. Instead of learning the score, we opt for directly learning the weighted score $Q\nabla_u \log p(u_t, t)$. This design choice leads to faster sampling since we do not need to perform any expensive matrix multiplication with $Q$ at inference time.

A generalized version of Tweedie's formula (for proof see Appendix A.2) implies:

$$Q\nabla_u \log p(u_t, t) = \frac{\mathbb{E}[u_0|u_t] - u_t}{\sigma^2(t)}. \tag{5}$$

We approximate $\mathbb{E}[u_0|u_t]$ with a neural network $h_\theta$ by minimizing the denoising objective:

$$\mathbb{E}_{t \sim U(0,\tau)}\mathbb{E}_{u_0 \sim \mu}\mathbb{E}_{u_t \sim N(u_0, \sigma^2(t)Q)}|h_\theta(u_t, t) - u_0|^2. \tag{6}$$

Having a minimizer $h_\theta$ of (6) gives us access to the weighted score $Q\nabla_u \log p(u_t, t)$ via (5). We may then obtain an approximate solution to the map $u(\tau) \mapsto u(0)$ by discretizing (4) in time. We consider Euler scheme updates given by

$$u_{t-\Delta t} = u_t - \Delta t \frac{\dot{\sigma}(t)}{\sigma(t)} \big( h_\theta(u_t, t) - u_t \big). \tag{7}$$

started with $u_\tau \sim N(0, \sigma^2(\tau)Q)$ for some time step $\Delta t > 0$.

**Equivariance for the Probability Flow ODE.** Since the diffusion model works with discrete inputs, we need to introduce a discretization of (2) for the network. For a deformation $T_1$, we define equivariance as

$$h_\theta(u_t \circ T_1^{-1}, t) \circ T_1 = h_\theta(u_t, t), \tag{8}$$

which is obtained from composing both sides of (2) with $T_1$. Note that (8) is valid only for pixels which stay within frame and we compute the l.h.s. with bilinear interpolation on the network output. The input to the network on the l.h.s. is computed with RFFs without any interpolation. Given this discrete equivariance is satisfied for every prediction of the network, it is straightforward to show that the whole diffusion model sampling chain will be equivariant. Indeed, the whole approximation to $u(\tau) \mapsto u(0)$ is equivariant by the linearity of composition – for a full derivation, see Appendix A.3.

**Equivariance Self-Guidance.** The condition (8) is rarely satisfied for deformations $T_1$ arising in practical settings. This is because either the conditional expectation $\mathbb{E}[u_0|u_t]$ is not equivariant with respect to $T_1^{-1}$ or the neural network approximation has not fully captured it. If the underlying equivariance assumption breaks, methods that rely solely on noise warping for temporal consistency, e.g. [14], will perform poorly. This is evident in challenging conditional tasks (see Figure 1).

A potential solution is to directly train the network by adding (8) as a regularizer. However, this requires large amounts of video data from which to extract optical flows. Furthermore, by satisfying (8) over a large class of $T_1$(s), the network may become less apt at satisfying (6) and lose its generative abilities. Therefore, we opt for *guiding the model towards equivariant solutions at inference time*.

We first sample noise $u_\tau^{(0)}$ and generate the first frame following (7), keeping the outputs of the network at each time step $\{h_\theta(u_t^{(0)}, t)\}$. To generative the next frame, we warp our noise $u_\tau^{(1)} = u_\tau^{(0)} \circ T_1^{-1}$ with RFFs and again follow (7) but this time using (8) as guidance. In particular, we take a gradient steps in the direction of the loss function $|h_\theta(u_t^{(1)}, t) \circ T_1 - h_\theta(u_t^{(0)}, t)|^2$, computed on the pixels that stay within frame. All frames can be generated by iterating this procedure as summarized in Algorithm 1 (and visualized in Figures 2, 9) which also shows how to use conditioning information. Guidance is typically used to solve inverse problems with diffusion models (e.g. see [15]), but here the guidance is applied to align the model with its own past predictions. We emphasize that to compute the composition with $T_1$ above, we use bilinear interpolation on the network outputs but we never need to interpolate the network inputs since we can compute the warping via RFFs. Furthermore, since we are matching interpolated outputs to ones that are not interpolated, our output images remain sharp. This in contrast to directly using a discrete version of (2) which would suggest that we match network outputs to interpolated images, producing blurry results.

## 4 Experimental Results

For all our experiments, we use Stable Diffusion XL [60] (SDXL) as our base image diffusion model. We start by finetuning SDXL on conditional tasks. We choose super-resolution and inpainting as the tasks of interest since they are both commonly used in the inverse problems literature and they represent two distinct scenarios: in super-resolution, the input condition is strong and in inpainting, the model needs to generate new content. For super-resolution, we choose a downsampling factor of $8$. For inpainting, we create masks of different shapes at random, following the work of [57]. During the finetuning, we train the model to predict the uncorrupted image given the following inputs: i) the encoding of the noised image, ii) the noise level, and, iii) the encoding of the corrupted (downsampled/masked) image. To condition on the corrupted observation, we concatenate the measurements across the channel dimension. We train models with and without correlated noise on the COYO dataset [12] for 100k steps. We show realizations of independent and correlated noise in Figure 6. Additional implementation details are in Section F.2, including the parameters for the GP introduced in Section 3.1.

---

**Algorithm 1** Warped Diffusion – Temporal Consistency with Equivariance Self Guidance

---

**Require:** Conditioning vectors $\{c_j\}_{j=0}^n$, Step $\Delta t$, Time $\tau$, Schedule $\sigma(t)$, Model $h_\theta$, Guidance Strength $\lambda$.

1: $\{T_j, T_j^{-1}\}_{j=1}^n \leftarrow$ `compute_optical_flow`$(\{c_j\}_{j=0}^n)$

2: $u_\tau^{(0)} \sim$ GP using RFFs in Section 3.1          ▷ Fresh noise sample for the first frame

3: Compute trajectory $\{u_t^{(0)}\}$ using (7)          ▷ Sample first frame

4: **for** $j \leftarrow 1$ to $n$ **do**

5:     $u_\tau^{(j)} \leftarrow u_\tau^{(j-1)} \circ T_j^{-1}$ using RFFs          ▷ Warp noise from previous frame

6:     $t \leftarrow \tau$

7:     **while** $t > 0$ **do**

8:         $u_{t-\Delta t}^{(j)} \leftarrow u_t^{(j)} - \Delta t \frac{\dot\sigma(t)}{\sigma(t)}\left(h_\theta(u_t^{(j)}, t, c_j) - u_t^{(j)}\right)$          ▷ Take Euler step

9:         $e_t^{(j)} \leftarrow \left| h_\theta(u_t^{(j)}, t, c_j) \circ T_j - h_\theta(u_t^{(j-1)}, t, c_{j-1}) \right|^2$          ▷ Compute warping error

10:         $u_{t-\Delta t}^{(j)} \leftarrow u_{t-\Delta t}^{(j)} - \frac{\lambda}{\sqrt{e_t}} \nabla_u e_t^{(j)}$          ▷ Equivariance self guidance

11:         $t \leftarrow t - \Delta t$

12:     **end while**

13: **end for**

14: **return** video $\{u_0^{(j)}\}_{j=0}^n$

---

## 4.1 Training with correlated noise

The first step is to assess the quality of the trained models. To do so, we take images from a test split of the COYO dataset, we corrupt them (either by masking or downsampling) and we measure the conditional performance of the trained models. We use a diverse set of metrics that are commonly used in the inverse problems literature: CLIP Text Score [62], CLIP Image Score [62], SSIM [85], LPIPS [91], MSE, Inception Score [76] and FID [37]. The first five metrics measure point-wise restoration performance. Inception Score measures the quality of the generated distribution (without an explicit reference distribution). Finally, FID measures restoration performance in a distributional sense, i.e. it measures how close is the distribution after restoration to the ground truth distribution.

We report our results for the super-resolution and inpainting models in Table 1. The main finding is that finetuning with correlated noise does not compromise performance, i.e. SDXL models finetuned with correlated noise perform on par with SDXL models that are trained with independent noise. Particularly for inpainting, the GP models slightly outperform models trained with independent noise across all metrics. We provide qualitative results for our all models in Figure 1 and in Appendix Figures 7, 8.

We remark that the advantages of using an initial distribution other than white noise have been explored in prior work [20, 6, 42]. Our new finding is that a model initially trained with white noise can be easily fine-tuned to work with correlated noise. To the best of our knowledge, ours is the first work that shows that Stable Diffusion XL can be fine-tuned to work with correlated noise.

We underline that prior to fine-tuning Stable Diffusion XL produces unrealistic images when the sampling chain is initialized with correlated noise. Our experiments show that post-finetuning, the model can handle spatially correlated noise in the input without compromising performance. Our GP Warping mechanism requires models that can handle correlated noise. Hence, these fine-tunings are essential for the rest of the paper.

Table 1: Single-frame evaluation of super-resolution and inpainting models.

| Model | FID ↓ | Inception ↑ | CLIP Txt ↑ | CLIP Img ↑ | SSIM ↑ | LPIPS ↓ | MSE ↓ |
|---|---|---|---|---|---|---|---|
| Super-resolution GP | **37.514** | **11.917** | **0.272**±**0.042** | 0.955±0.029 | 0.770±0.106 | 0.253±0.076 | 0.004±0.004 |
| Super-resolution Indep. | 40.843 | 11.679 | 0.271±0.041 | **0.957**±**0.027** | **0.785**±**0.108** | **0.242**±**0.078** | 0.004±0.004 |
| Inpainting GP | **58.727** | **11.769** | **0.276**±**0.042** | **0.929**±**0.060** | **0.798**±**0.134** | **0.181**±**0.122** | **0.056**±**0.084** |
| Inpainting Indep. | 61.380 | 11.707 | 0.275±0.048 | 0.913±0.089 | 0.778±0.161 | 0.198±0.134 | 0.057±0.076 |

## 4.2 Noise Warping and Equivariance Self Guidance

In the previous experiments, we measured the restoration performance of the trained models for a single image and we established that models trained with correlated noise perform on par (or even outperform) models trained with independent noise. The next step is to measure the temporal behavior of the models, i.e. how well they work for videos.

**Noise Warping baselines.** As explained in Section 1, to apply image diffusion models to videos, we need to transform the noise as we move from one frame to the next. We consider the following noise-warping baselines that were used in [14]: **Fixed noise** uses the same noise across all the frames. **Resample noise** samples a new noise for each new frame. **Nearest Neighbor** uses the noise of the nearest location in the grid to evaluate the noise at the location that is not on the regular grid $E_k$. **Bilinear Interpolation** interpolates the values of the noise bilinearly in the neighboring locations that lie on the grid. **How I Warped Your Noise [14]** is the state-of-the-art method for solving temporally correlated inverse problems with image diffusion models. It warps the noise by using auxiliary high-resolution noise maps (see our intro, related work section, and Section C). **Our GP Noise Warping** warps the input noise by resampling the Gaussian process in the mapped locations. We note that the Fixed Noise, Resample Noise, and Nearest Neighbor noise warping methods can be applied to models that are trained with either independent noise or correlated noise coming from a GP. For all the experiments, we also include our proposed method, *Warped Diffusion* that uses GP Noise Warping and Equivariance Self-Guidance (see Algorithm 1 for a reference implementation).

**Video Evaluation Metrics.** We follow the evaluation methodology of the "How I Warped Your Noise" paper [14]. Specifically, we want to measure two different aspects of our method: i) average restoration performance across frames, ii) temporal consistency. For i), we measure the average of all the previously reported metrics (FID, Inception, CLIP Image/Text score, SSIM, LPIPS and MSE) across the frames. For ii), we measure the self-warping error, i.e. how consistent are the model's predictions across time. The warping error can be computed in either pixel or latent space and also with respect to the first generated frame or the previously generated frame, totaling 4 warping errors.

To warm up, we start with videos that are synthetically generated by 2-D shifting of a single image, as in Figure 1. To further simplify the setup, we consider the easy case of shifting the current frame by an integer amount of pixels with each new frame. For 2-D translations by an integer amount of pixels, the Nearest Neighbor, Bilinear Interpolation, How I Warped Your Noise and GP Noise Warping methods they become essentially the same since we always evaluate the noise distribution on points in the grid $E_k$. Hence, the only difference is whether we apply these methods to white noise or to GPs.

Figure 1 (Row 2) shows that the How I Warped Your Noise baseline produces temporally inconsistent results as we shift the masked input image. Even though all the inpaintings are of high quality, the baseline results are temporally inconsistent. Instead, our *Warped Diffusion* method produces temporally consistent results since it enforces equivariance by design. Since the How I Warped Your Noise warping mechanism and GP coincide here, the benefit strictly comes from enforcing the equivariance property. In fact, one could get the same results for the How I Warped Your Noise method by penalizing for equivariance at inference time.

We present quantitative results regarding temporal consistency in Figure 3 (and additional results in Figure 4 in the Appendix). As shown in the Figure, the fixed noise and the resample noise baselines perform the worst w.r.t. the temporal consistency both in latent and pixel space. The warping error of the Resample baseline is almost constant across frames as expected, while the warping error of the Fixed Noise increases with time. Both the How I Warped Your Noise method and our GP warping framework significantly improve the baselines. Yet, they still have significant temporal inconsistencies as evidenced by the results in Figure 1 and the supplemental videos. The two methods perform on par on this task since they are essentially the same when it comes to integer shifts: the only difference is that GP Noise Warping is applied to correlated noise coming from a GP. The remaining temporal errors are not an artifact of the noise warping mechanism but they are due to the fact that the model itself is not equivariant w.r.t. the underlying transformation. The warping errors essentially disappear when we apply Equivariance Self Guidance. As shown in Figure 3, our method, Warped Diffusion, achieves almost 0 warping error (1e-4 mean pixel error with respect to the first frame to be precise) since it is enforcing equivariance by design.

The only remaining question is whether Warped Diffusion maintains good restoration performance. To answer this, we measure mean restoration performance across frames for the aforementioned metrics. We report our results in Table 2, including the mean warping error with respect to the first frame. As shown, Warped Diffusion maintains high performance across all the considered metrics while being significantly superior in terms of temporal consistency. The conclusion is that all the other noise warping baselines, including the previous state-of-the-art How I Warped Your Noise paper [14], perform poorly in terms of temporal consistency since they rely on the assumption that the network is equivariant. Even for simple temporal correlations such as integer movement in the 2-D space, this

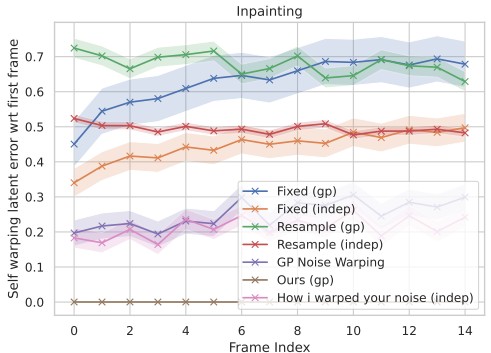
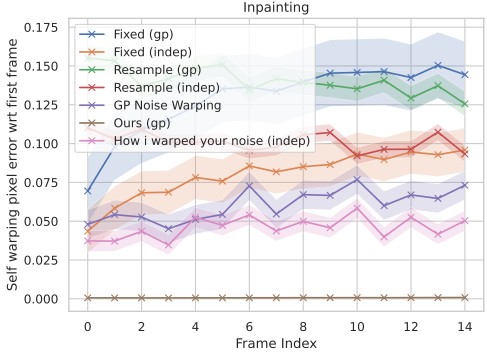

(a) Self-warping error w.r.t. first frame in latent space.     (b) Self-warping error w.r.t. first frame in pixel space.

Figure 3: Self-warping error w.r.t. first frame for the inpainting task as we shift the input frame.

assumption is false for the challenging inpainting task. Warped Diffusion is the only method that achieves temporal consistency while it still manages to maintain high reconstruction performance.

Table 2: Mean-frame evaluation of inpainting models for the translation task.

| Method | Warping Err ↓ | FID ↓ | Inception ↑ | CLIP Txt ↑ | CLIP Img ↑ | SSIM ↑ | LPIPS ↓ | MSE ↓ |
|---|---|---|---|---|---|---|---|---|
| Fixed (gp) | $0.129_{\pm 0.022}$ | $60.853_{\pm 2.908}$ | $\mathbf{12.421_{\pm 0.761}}$ | $\mathbf{0.280_{\pm 0.003}}$ | $0.924_{\pm 0.005}$ | $0.800_{\pm 0.001}$ | $\mathbf{0.182_{\pm 0.001}}$ | $0.060_{\pm 0.002}$ |
| Fixed (indep) | $0.080_{\pm 0.014}$ | $67.021_{\pm 2.696}$ | $10.301_{\pm 0.392}$ | $0.275_{\pm 0.002}$ | $0.919_{\pm 0.004}$ | $0.780_{\pm 0.001}$ | $0.195_{\pm 0.001}$ | $0.059_{\pm 0.002}$ |
| Resample (indep) | $0.101_{\pm 0.006}$ | $71.078_{\pm 4.185}$ | $11.740_{\pm 0.435}$ | $0.277_{\pm 0.002}$ | $0.921_{\pm 0.004}$ | $0.781_{\pm 0.006}$ | $0.196_{\pm 0.002}$ | $0.061_{\pm 0.003}$ |
| Resample (gp) | $0.141_{\pm 0.008}$ | $60.029_{\pm 4.389}$ | $11.318_{\pm 0.403}$ | $0.277_{\pm 0.002}$ | $\mathbf{0.925_{\pm 0.003}}$ | $\mathbf{0.806_{\pm 0.005}}$ | $\mathbf{0.182_{\pm 0.002}}$ | $\mathbf{0.056_{\pm 0.003}}$ |
| How I Warped (indep) | $0.046_{\pm 0.007}$ | $68.701_{\pm 2.938}$ | $10.877_{\pm 0.432}$ | $0.276_{\pm 0.001}$ | $0.910_{\pm 0.005}$ | $0.781_{\pm 0.001}$ | $0.197_{\pm 0.001}$ | $0.067_{\pm 0.001}$ |
| GP Warping | $0.061_{\pm 0.010}$ | $\mathbf{59.897_{\pm 3.718}}$ | $11.727_{\pm 0.375}$ | $0.277_{\pm 0.002}$ | $0.924_{\pm 0.004}$ | $0.803_{\pm 0.002}$ | $\mathbf{0.182_{\pm 0.001}}$ | $0.057_{\pm 0.002}$ |
| **Warped Diffusion** (Ours) | $\mathbf{0.001_{\pm 0.001}}$ | $61.249_{\pm 2.499}$ | $11.802_{\pm 0.427}$ | $0.276_{\pm 0.001}$ | $0.917_{\pm 0.006}$ | $0.779_{\pm 0.011}$ | $0.188_{\pm 0.006}$ | $0.058_{\pm 0.001}$ |

We finally remark that our sampling algorithm enforces equivariance in the latent space. Yet, the warping errors are negligible in the pixel space as well. Our finding is that improving latent space equivariance translates to improvements in pixel space equivariance. The authors of [14] also find that "the VAE decoder is translationally equivariant in a discrete way".

### 4.3   Effect of Sampling Guidance for more general transformations

We proceed to evaluate our method on realistic videos. We measure performance on 600 captioned videos from the FETV [55] dataset. Since baseline inpainting methods fail even for very simple temporal transformations, we focus on $8\times$ super-resolution for our comparisons on FETV.

For our video results, we could not provide comparisons with the How I Warped Your Noise paper. At the time of this writing, there was no available reference implementation as we confirmed with the authors by direct communication. In any case, the authors acknowledge as a limitation of their work that their proposed method has "limited impact on temporal coherency" when applied to latent models and that "all the noise schemes produce temporally inconsistent results" [14]. Once again, we attribute this to the non-equivariance of the denoiser, which we mitigate with our guidance algorithm.

We proceed to evaluate our method and the baselines with respect to temporal consistency and mean restoration performance across frames, as we did for our inpainting experiments. We present our results in Table 3 and additional results in Figures 5, 8 of the Appendix and in the following URL as videos: https://giannisdaras.github.io/warped_diffusion.github.io/. As shown in Table 3, there is a trade-off between temporal consistency and restoration performance. Methods that perform better in terms of temporal consistency often have significantly worse performance across the other metrics. Our Warped Diffusion achieves a sweet spot: it has the lowest warping error by a large margin and it still maintains competitive performance across all the other metrics. On the contrary, methods that are based solely on noise warping, such as GP Warping and the simple interpolation methods, lead to significant performance deterioration for a small improvement in temporal consistency.

**Noise Warping Speed.** We measure the time needed for a single noise warping. Our GP Warping mechanism takes 39ms per frame Wall Clock time, to produce the warping at $1024 \times 1024$ resolution. This is $16\times$ faster than the reported 629ms number in [14]. If we use batch parallelization, our method generates 1000 noise warpings in just 46ms (at the expense of extra memory).

**No Warping?** A natural question is whether we can omit completely the noise warping scheme since equivariance is forced at inference time. We ran some preliminary experiments for super-

Table 3: Mean-frame evaluation of super-resolution models for real videos.

| Method | Warping Err ↓ | FID ↓ | Inception ↑ | CLIP Txt ↑ | CLIP Img ↑ | SSIM ↑ | LPIPS ↓ | MSE ↓ |
|---|---|---|---|---|---|---|---|---|
| Fixed (indep) | $0.940_{\pm 0.312}$ | $48.764_{\pm 2.592}$ | $8.746_{\pm 1.325}$ | $0.227_{\pm 0.002}$ | $0.948_{\pm 0.013}$ | $\mathbf{0.716}_{\pm \mathbf{0.023}}$ | $0.188_{\pm 0.018}$ | $0.005_{\pm 0.001}$ |
| Resample (indep) | $0.934_{\pm 0.341}$ | $\mathbf{47.550}_{\pm \mathbf{2.434}}$ | $8.879_{\pm 1.337}$ | $0.229_{\pm 0.002}$ | $0.948_{\pm 0.011}$ | $0.708_{\pm 0.021}$ | $\mathbf{0.183}_{\pm \mathbf{0.018}}$ | $\mathbf{0.005}_{\pm \mathbf{0.001}}$ |
| Nearest (indep) | $1.048_{\pm 0.381}$ | $67.078_{\pm 6.359}$ | $8.608_{\pm 1.339}$ | $0.228_{\pm 0.002}$ | $0.943_{\pm 0.009}$ | $0.683_{\pm 0.022}$ | $0.227_{\pm 0.031}$ | $0.007_{\pm 0.002}$ |
| Bilinear (indep) | $0.990_{\pm 0.372}$ | $66.330_{\pm 6.394}$ | $8.832_{\pm 1.480}$ | $0.228_{\pm 0.002}$ | $0.942_{\pm 0.012}$ | $0.684_{\pm 0.019}$ | $0.216_{\pm 0.029}$ | $0.008_{\pm 0.002}$ |
| Fixed (gp) | $1.006_{\pm 0.362}$ | $54.058_{\pm 4.299}$ | $8.045_{\pm 1.205}$ | $0.222_{\pm 0.002}$ | $0.954_{\pm 0.007}$ | $0.666_{\pm 0.019}$ | $0.198_{\pm 0.021}$ | $0.007_{\pm 0.001}$ |
| Resample (gp) | $0.974_{\pm 0.308}$ | $54.778_{\pm 3.942}$ | $\mathbf{9.471}_{\pm \mathbf{1.566}}$ | $0.225_{\pm 0.004}$ | $\mathbf{0.954}_{\pm \mathbf{0.007}}$ | $0.661_{\pm 0.025}$ | $0.209_{\pm 0.020}$ | $0.006_{\pm 0.001}$ |
| Nearest (gp) | $0.975_{\pm 0.383}$ | $79.743_{\pm 10.835}$ | $8.896_{\pm 1.462}$ | $0.224_{\pm 0.002}$ | $0.939_{\pm 0.004}$ | $0.637_{\pm 0.015}$ | $0.243_{\pm 0.044}$ | $0.009_{\pm 0.003}$ |
| Bilinear (gp) | $0.953_{\pm 0.390}$ | $78.866_{\pm 11.960}$ | $8.565_{\pm 1.537}$ | $0.228_{\pm 0.001}$ | $0.942_{\pm 0.005}$ | $0.635_{\pm 0.014}$ | $0.247_{\pm 0.046}$ | $0.009_{\pm 0.003}$ |
| GP Warping | $0.812_{\pm 0.337}$ | $75.763_{\pm 11.555}$ | $8.291_{\pm 1.168}$ | $0.225_{\pm 0.002}$ | $0.941_{\pm 0.006}$ | $0.653_{\pm 0.016}$ | $0.226_{\pm 0.043}$ | $0.008_{\pm 0.003}$ |
| **Warped Diffusion** (Ours) | $\mathbf{0.649}_{\pm \mathbf{0.363}}$ | $58.189_{\pm 6.322}$ | $8.882_{\pm 1.704}$ | $\mathbf{0.235}_{\pm \mathbf{0.003}}$ | $0.943_{\pm 0.005}$ | $0.654_{\pm 0.024}$ | $0.221_{\pm 0.041}$ | $0.008_{\pm 0.002}$ |

resolution on real-videos and we found that omitting the warping significantly deteriorates the results when the number of sampling steps is low. We found that increasing the number of sampling steps makes the effect of the initial noise warping less significant, at the cost of increased sampling time.

## 5   Limitations

Our method has several limitations. First, the guidance term increases the sampling time, as detailed in the Appendix, Section F.3. For reference, processing a 2-second video takes roughly 5 minutes on a single A-100 GPU. Second, even though in our experiments we observed a monotonic relation between the warping error in latent space and warping error in pixel space, it is possible that for some transformations the decoder of a Latent Diffusion Model might not be equivariant. We noticed that this is a common failure for text rendering, e.g. in this latent video the model seems to be equivariant, but in the pixel video it is not. Third, the success of our method depends on the quality of the flow estimation – inconsistent flow estimation between frames will lead to flickering artifacts. For real videos, there might be occlusions and the estimation of the flow map can be noisy. We observed that in such cases our method fails, especially for challenging tasks such as video inpainting. The correlations obtained by following the optical flow field obtained from real videos might lead to a distribution shift compared to the training distribution. For such extreme deformations, our method produces correlation artifacts. This has been observed in prior work (see this video), but it also appears in our setting (e.g. see this video). Finally, our method cannot work in a zero-shot manner since it requires a model that is trained with correlated noise.

## 6   Conclusions

Warped Diffusion is a novel framework for solving temporally correlated inverse problems with image diffusion models. It leverages a noise warping scheme based on Gaussian processes to propagate noise maps and it ensures equivariant generation through an efficient equivariance self-guidance technique. We extensively validated Warped Diffusion on temporally coherent inpainting and superresolution, where our approach outperforms relevant baselines both quantitatively and qualitatively. Importantly, in contrast to previous work [14], our method can be applied seamlessly also to *latent* diffusion models, including state-of-the-art text-to-image models like SDXL [60].

## 7   Acknowledgements

This research has been partially supported by NSF Grants AF 1901292, CNS 2148141, Tripods CCF 1934932, IFML CCF 2019844 and research gifts by Western Digital, Amazon, WNCG IAP, UT Austin Machine Learning Lab (MLL), Cisco and the Stanly P. Finch Centennial Professorship in Engineering. Giannis Daras has been partially supported by the Onassis Fellowship (Scholarship ID: F ZS 012-1/2022-2023), the Bodossaki Fellowship and the Leventis Fellowship.

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

## A    Theoretical Results

### A.1    Convolutions and Equivariance

To better understand (2), we consider the following example. We will assume that $\xi : \mathbb{R}^2 \to \mathbb{R}$ is a scalar-valued field (grayscale image) defined on the whole plane. Furthermore, we let $G$ be given by a continuous convolution and $T_1^{-1}$ be a translation. In particular, for all $x \in \mathbb{R}^2$,

$$G(\xi)(x) = \int_{\mathbb{R}^2} \kappa(x - y)\xi(y) \, \mathsf{d}y, \qquad T_1^{-1}(x) = x - a$$

for some compactly supported kernel $\kappa : \mathbb{R}^2 \to \mathbb{R}$ and a direction $a \in \mathbb{R}^2$. We then have

$$G\big(\xi \circ T_1^{-1}\big)(x) = \int_{\mathbb{R}^2} \kappa(x - y)\xi(y - a) \, \mathsf{d}y = \int_{\mathbb{R}^2} \kappa\big((x - a) - y\big)\xi(y) \, \mathsf{d}y = G\big(\xi\big)\big(T_1^{-1}(x)\big)$$

by the change of variables formula. This shows that $G$ is equivariant to all translations. This is a well-known property of the convolution and, in particular, it shows that convolutional neural networks

are translation equivariant, noting that pointwise non-linearities will preserve this property. This example shows that a model can be equivariant with respect to certain deformations by architectural design. However, in realstic video modeling, optical flows are not know explicitly and can only be approximated numerically. It is therefore natural to instead build-in approximate equivariance into a model instead of enforcing it directly in the architecture. Our guidance procedure in Section 3.2 is an example such an approximate form of equivariance.

## A.2 Tweedie's Formula

In Section 3.2, we show a diffusion model can be trained and sampled from using Gaussian process noise instead of white noise. Our result depends on the following lemma which is a simple generalization of Tweedie's formula.

**Lemma A.1.** *Let $x$ be a random variable with positive density $p_x \in C^1(\mathbb{R}^k)$. Let $\sigma > 0$ and $z \sim \mathcal{N}(0, Q)$ for some positive definite matrix $Q \in \mathbb{R}^{k \times k}$ and assume that $x \perp z$. Define the random variable*

$$y = x + \sigma z$$

*and let $p_y \in C^\infty(\mathbb{R}^k)$ be the density of $y$. It holds that*

$$\nabla_y \log p_y(y) = \frac{1}{\sigma^2} Q^{-1} \big( \mathbb{E}[x|y] - y \big).$$

*Proof.* First note that by the chain rule,

$$\nabla_y \log p_y(y) = \frac{1}{p_y(y)} \nabla_y p_y(y) = \frac{1}{p_y(y)} \nabla_y \int_{\mathbb{R}^k} p(y, x) \, \mathrm{d}x$$

where $p(y, x)$ denotes the joint density of $(y, x)$. Let $p(y|x)$ denote the Gaussian density of the conditional $y|x$. Since $p_x \in C^1(\mathbb{R}^k)$,

$$\nabla_y \int_{\mathbb{R}^k} p(y, x) \, \mathrm{d}x = \int_{\mathbb{R}^k} \nabla_y p(y|x) p_x(x) \, \mathrm{d}x.$$

Therefore, by the chain rule,

$$\nabla_y \log p_y(y) = \frac{1}{p_y(y)} \int_{\mathbb{R}^k} p(y|x) p_x(x) \nabla_y \log p(y|x) \, \mathrm{d}x.$$

Since $p(y|x)$ is the density of $\mathcal{N}(x, \sigma^2 Q)$, a direct calculations shows that

$$\nabla_y \log p(y|x) = \frac{1}{\sigma^2} Q^{-1} \big( x - y \big).$$

Furthermore, Bayes' theorem implies

$$p(y|x) p_x(x) = p(x|y) p_y(y).$$

Therefore,

$$\nabla_y \log p_y(y) = \frac{1}{\sigma^2} \int_{\mathbb{R}^k} Q^{-1} \big( x - y \big) p(x|y) \, \mathrm{d}x = \frac{1}{\sigma^2} Q^{-1} \big( \mathbb{E}[x|y] - y \big)$$

as desired. □

## A.3 Flow Equivariance

In Section 3.2, we claim that if the score network $h_\theta$ is equivarient with respect to a deformation $T^{-1}$, then the Euler scheme approximation of the map $u(\tau) \mapsto u(0)$ is equivariant with respect to $T^{-1}$. It is easy to see that this results holds so long as it holds for the single step $u_t \mapsto u_{t-\Delta t}$ defined by (7). We will assume that $h_\theta$ safisfies (8) written as

$$h_\theta(u_t \circ T^{-1}, t) = h_\theta(u_t, t) \circ T^{-1}.$$

We make sense of this equation by using RFF to define $u_t$ as a function on the plane and similarly bilinear interpolation to define $h_\theta(u_t, t)$ as a function. It follows by linearity of composition that

$$u_{t-\Delta t} \circ T^{-1} = u_t \circ T^{-1} - \Delta t \frac{\dot{\sigma}(t)}{\sigma(t)} \big( h_\theta(u_t, t) \circ T^{-1} - u_t \circ T^{-1} \big)$$

$$= u_t \circ T^{-1} - \Delta t \frac{\dot{\sigma}(t)}{\sigma(t)} \big( h_\theta(u_t \circ T^{-1}, t) - u_t \circ T^{-1} \big)$$

which is the requisite equivariance of the map $u_t \mapsto u_{t-\Delta t}$.

## B   Gaussian Processes

A probability measure $\eta$ on $H$ is called Gaussian if there exists an element $m \in H$ and a self-adjoint, non-negative, trace-class operator $Q : H \to H$ such that, for all $h, h' \in H$,

$$\langle h, m \rangle = \int_H \langle h, f \rangle \, \mathrm{d}\eta(f), \quad \langle Qh, h' \rangle = \int_H \langle h, f - m \rangle \langle h', f - m \rangle \, \mathrm{d}\eta(f),$$

where $\langle \cdot, \cdot \rangle$ denotes the inner product on $H$. The element $m$ is called the *mean* while the operator $Q$ is called the *covariance*. It is immediate from this definition that white noise is not included since the identity operator is not trace-class on any infinite dimensional space. This definition ensures that any realization of a random variable $\xi \sim \eta$ is almost surely an element of $H$. When the domain $D$ of the elements of $H$ is a subset of the real line, $\eta$ is often called a *Gaussian process*. We continue to use this terminology even when $D$ is a subset of a higher dimensional space i.e. $\mathbb{R}^2$ but remark that the nomenclature *Gaussian random field* is sometimes preferred.

Since we working on a separable space, each such field on $H$ has associated to it a unique reproducing kernel Hilbert space [18, Theorem 2.9] which is associated to a unique positive definite kernel [4]. In particular, there exists a positive definite function $\kappa : D \times D \to \mathbb{R}$ for which $Q$ is its associated integral operator. It follows that a Gaussian process can be uniquely identified with a positive definite kernel. Sampling and conditioning this process can then be accomplished via the kernel matrix.

To make this explicit, suppose that $X = \{x_1, \ldots, x_n\} \subset D$ and $Y = \{y_1, \ldots, y_m\} \subset D$ are two sets of points in $D$. We will slightly abuse notation and write

$$Q(X, Y)_{ij} := \kappa(x_i, y_j), \qquad i = 1, \ldots, n \text{ and } j = 1, \ldots, m$$

for the kernel matrix between $X$ and $Y$ and similarly $Q(Y, X), Q(X, X), Q(Y, Y)$. Suppose that $\xi \sim \eta$ is a random variable from the Gaussian process with kernel $\kappa$ and mean zero. To sample a realization of $\xi$ on the points $X$, we sample the finite dimensional Gaussian $\mathcal{N}\big(0, Q(X, X)\big)$. This can be written as

$$\xi(X) = Q(X, X)^{1/2} Z$$

where $Z \sim \mathcal{N}(0, I_n)$. Suppose now that the points in $Y$ are distinct from those in $X$ and we want to sample $\xi$ on $Y$ given the realization $\xi(X)$. This can be done by conditioning [65]

$$\xi(Y) | \xi(X) \sim \mathcal{N}\big(Q(Y, X) Q(X, X)^{-1} \xi(X), Q(Y, Y) - Q(Y, X) Q(X, X)^{-1} Q(X, Y)\big).$$

While the above formulas fully characterize sampling $\xi$, working with them can be computationally burdensome. It is therefore of interest to consider a different viewpoint on Gaussian processes, in particular, through the Karhunen–Loève expansion. The spectral theorem implies that $Q$ possesses a full set of eigenfunctions $Q_j \phi_j = \lambda_j \phi_j$ for $j = 1, 2, \ldots$ with some decaying sequence of eigenvalues $\lambda_1 \geq \lambda_2 \geq \ldots$. The random variable $\xi \sim \mathcal{N}(0, Q)$ can be written as

$$\xi = \sum_{j=1}^{\infty} \sqrt{\lambda_j} \chi_j \phi_j$$

where $\chi_j \sim \mathcal{N}(0, 1)$ is an i.i.d. sequence and the right hand side sum converges almost surely in the norm of $H$ [18]. By truncating this sum to a finite number of terms, computing realizations of $\xi$ becomes much more computationally manageable. This inspires the random features approach to Gaussian processes which is the basis of our computational method; for precise details, see [65, 63, 87].

## C   Brownian Bridge Interpolation

We show in Section 2.3 that a white noise process is not compatible with the idea of using a generative model to interpolate deformed functions. A potential way of dealing with this issue is to treat the original realizations of the white noise $\xi(E_k)$ as the fixed nodal points of a function $\xi$ and obtain the rest of the values via interpolation. It is shown in [14] that common forms of interpolation yield a conditional distribution $\xi\big(T^{-1}(E_k)\big) | \xi(E_k)$ that is too dissimilar from the training distribution $\mathcal{N}(0, I_k)$ and thus the generative model produces blurry or disfigured images.

Therefore [14] proposes a stochastic interpolation method which has the property that, for a new point $x^* \notin E_k$, the distribution of $\xi(x^*)$ marginalized over the joint distribution $\big(\xi(E_k), \xi(x^*)\big)$ follows $\mathcal{N}(0,1)$. This is most easily seen in one spatial dimension with $k = 2$ points. Suppose that $D = [0,1]$ and let $a, b \sim \mathcal{N}(0,1)$ be two independent random variables. Consider a Gaussian process on $D$ with kernel function $\kappa(x, y) = 1 - |x - y|$ and suppose that $\xi$ is distributed according to this GP conditioned on $\xi(0) = a$ and $\xi(1) = b$. A straightforward calculation shows that, for any $x^* \in (0,1)$,

$$\xi(x^*) = (1 - x^*)a + x^*b + \sqrt{2x^*(1 - x^*)}z$$

for $z \sim \mathcal{N}(0,1)$ independent of $(a, b)$. This is simply the Brownian bridge connecting $a$ to $b$. Remarkably, the marginal distribution of $\xi(x^*)$ over the joint $(\xi(x^*), a, b)$ is $\mathcal{N}(0,1)$ independently of $x^*$. However, the conditional distribution is

$$\xi(x^*)|a, b = \mathcal{N}\big((1 - x^*)a + x^*b, 2x^*(1 - x^*)\big)$$

which is not $\mathcal{N}(0,1)$ for all $x^* \in (0,1)$. In [14, Section 2.2], it is proposed that such Brownian bridges are used between any two pair of pixels, yielding a stochastic interpolation method given by a sequence of such independent GPs. However, from the point of view of using a generative model that is pre-trained on $\mathcal{N}(0, I_k)$, it is not of interest that the marginal distribution of $\xi(x^*)$ is $\mathcal{N}(0,1)$ but rather that the conditional $\xi(x^*)|a, b$ is $\mathcal{N}(0,1)$. As we have seen, this is not the case for the method of [14] and, in fact, it will only ever be the case for white noise processes as discussed in Section 2.3. Therefore, no matter what method is used, there will always be a distribution shift to the model input induced by the deformation $T^{-1}$. A well chosen noise process will simply try to minimize this shift as much a possible.

The work [14] proposes to use diffusion models trained on discrete inputs distributed according to $\mathcal{N}(0, I_k)$ and computes conditional distributions $\xi\big(T^{-1}(E_k)\big)|\xi(E_k)$ using the stochastic interpolation method described above, generalized to two dimensions. We, instead, propose to use a Gaussian process $\mathcal{N}(0, Q)$, as described in Section 3.1 and compute $\xi\big(T^{-1}(E_k)\big)|\xi(E_k)$ by conditioning this process which amounts to simply evaluating the RFF projection. It is our numerical experience that this better preservers the qualitative properties of the input distribution for large deformations. We leave the exploration of a process best suited for this task as important future work.

## D   Additional Results

In this section, we provide additional results that did not fit in the main paper. We visualize the difference between independent noise and noise from our GP in Figure 6. We present inpainting results from our SDXL inpainting model fine-tuned with GP noise in Figure 7. We present super-resolution results from our SDXL super-resolution model fine-tuned with GP noise in Figure 8. We further present warping errors with respect to the previous frame in Figure 4 for the inpainting results and warping errors for super-resolution for real videos in Figure 5. Finally, we present additional comparisons for super-resolution in Figure 10.

## E   Related Works

Our work is primarily related to three recent lines of research about the utility of diffusion models in inverse problems, video editing, and equivariance in function space diffusion models as elaborated below.

**Diffusion Models for Inverse Problems.** Diffusion models have been recently received widespread adoption for solving inverse problems in various domains. Diffusion models can solve inverse problems in a few different ways. A simple way is to train or finetune a *conditional* diffusion model for each specific task to learn the conditional distribution from the degraded data distribution to the clean data distribution [71, 53, 77]. Some popular examples include SR3 [75] and inpainting stable diffusion [67]. We leverage stable diffusion inpainting in the present work. While successful, they however need to be trained (or finetuned) separately for each individual task that is computationally complex. Also, they are not robust to out of distribution data. To mitigate these challenges, *plug-and-play* methods have been introduced that utilize a single foundation diffusion model (e.g., stable diffusion) as a (rich) prior to solve many inverse problems at once [44, 70, 15, 47]. The crux of this approach is to modify the sampling post-hoc by either: $(i)$ add guidance to the score function

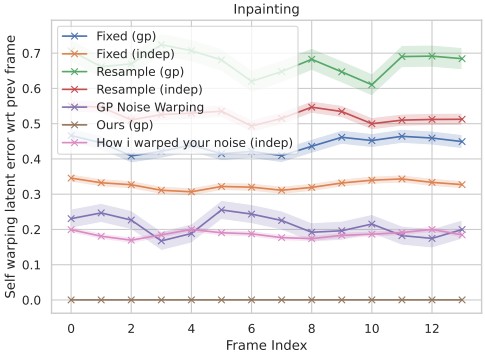
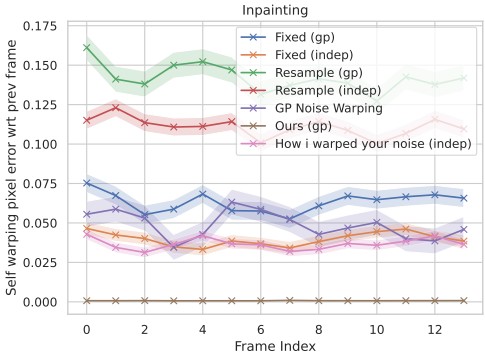

(a) Warping error w.r.t. previously generated frame in latent space.

(b) Warping error w.r.t. previously generated frame in pixel space.

Figure 4: Warping errors w.r.t. previously generated frame in latent and pixel space for the inpainting task as we shift the input frame.

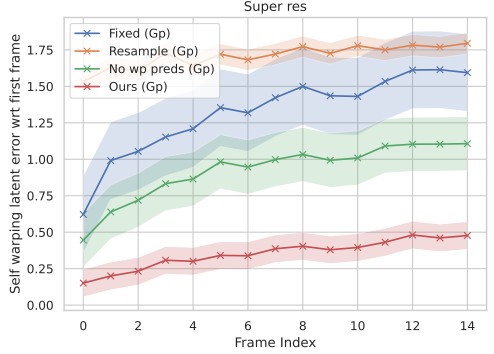
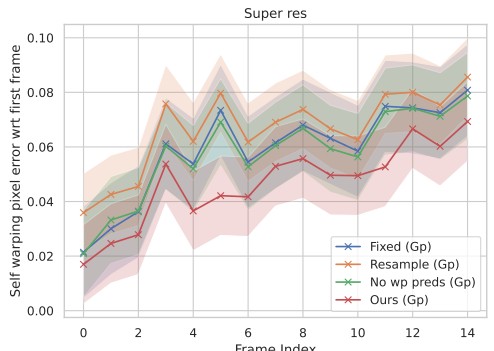

(a) Warping error w.r.t. first generated frame in latent space.

(b) Warping error w.r.t. first generated frame in pixel space.

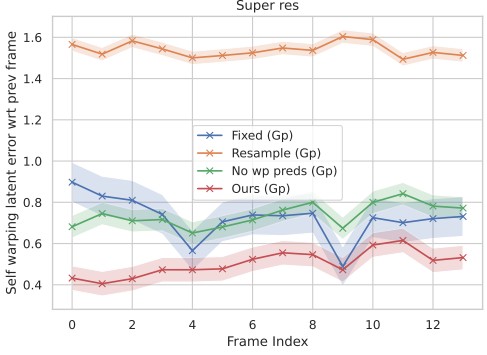
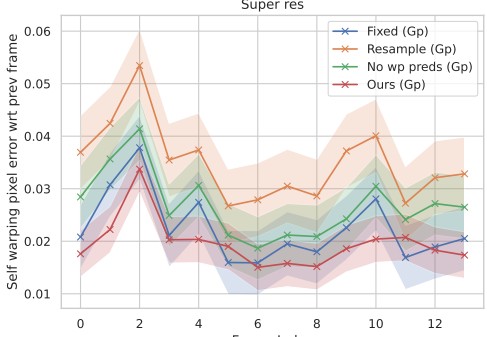

(c) Warping error w.r.t. previously generated frame in latent space.

(d) Warping error w.r.t. previously generated frame in pixel space.

Figure 5: Warping errors w.r.t. first generated frame (top-row) and prev. generated frame (bottom row) for the $8\times$ super-resolution task for real videos.

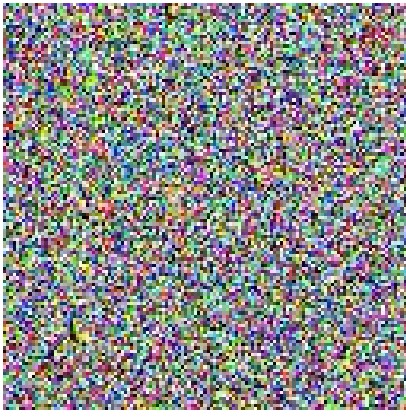

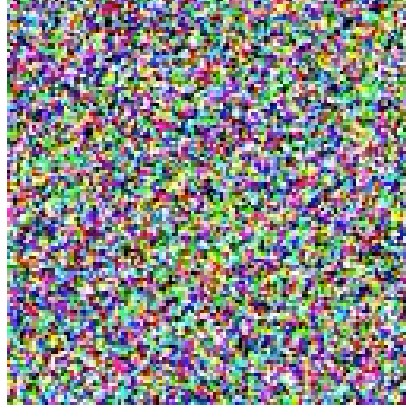

(a) Independent noise realization.    (b) Gaussian Process noise realization.

Figure 6: Visualization of independent noise and noise from a Gaussian Process.

of diffusion models as in [15, 81]; $(ii)$ approximated projection onto the measurement subspace at each diffusion step [16, 47] or, $(iii)$ use regularization by denoising via optimization [56, 32]. In this work we adopt the guidance-based approach to impose equivariance for the score function. All these methods have been applied for 2D images. For *video* inverse problems, the problem is more challenging due to temporal consistency. There are some efforts to leverage diffusion models for example for text-to-video superresolution or inpainting; see e.g., [69, 31, 93]. However, there is no systematic framework yet based on 2D diffusion models to solve generic video inverse problems in a temporally consistent manner. This is essentially the focus of our work. Finally, we remark that recent work [22, 21, 23, 2, 48, 1] has shown that it is even possible to train diffusion models to solve inverse problems without ever seeing clean images from the distribution of interest.

**Video Editing with Image Diffusion Models.** Due to the lack of full-fledged pre-trained text-to-video diffusion models, many works focus on video editing (or video-to-video translation) using text-to-image diffusion models. One line of research has proposed to fine-tune the image diffusion model on a single text-video pair and generate novel videos that represent the edits at inference [88, 54, 92]. Specifically, Tune-A-Video [88] proposed a cross-frame attention mechanism and an efficient one-shot tuning strategy. Video-P2P [54] further improved the video inversion performance by optimizing a shared unconditional embedding for all frames. EI$^2$ [92] refined the temporal modules to resolve semantic disparity and temporal inconsistency of video editing. However, the fine-tuning process over the input video makes the editing less efficient. Another line of research has developed various training-free methods for efficient video editing, which mostly rely on the cross-frame attention and latent fusion for maintaining temporal consistency [13, 51, 61, 90]. In particular, Text2Video-Zero [51] encoded the motion dynamics in latent noises through a noise wrapping. FateZero [61] fused the attention features with a blending mask obtained by the source prompt's cross-attention map. Pix2Video [13] proposed to progressively propagate the changes to the future frames via self-attention feature injection. Rerender-A-Video [90] proposed hierarchical cross-frame constraints with the optical flow for improved temporal consistency.

**Function Space Diffusion Models and Equivariance** Recently, several works [52, 49, 50] have extended diffusion models to function data. However, these methods primarily focus on theoretical developments and have been examined on simplistic datasets such as time series, Navier-Stokes solutions, or hand-written digits. This paper can be considered one of the first successful applications of function-space diffusion models to natural image datasets. Our work is also related to the equivariant diffusion models which have been extensively explored in scientific applications such as molecule and protein interaction and generation applications [43, 3, 89, 17, 45]. However, equivariant diffusion models for image generation are less explored, primarily because guaranteeing equivariance (for example with respect to translation, rotation, or rescaling) in commonly used diffusion architectures such as U-Net [68, 39] or Transformer [58, 36] models is challenging.

**Diffusion models trained with correlated noise.** Ours is not the first work to train diffusion models with a prior other than white noise. The authors of [20, 42] show how to train diffusion models

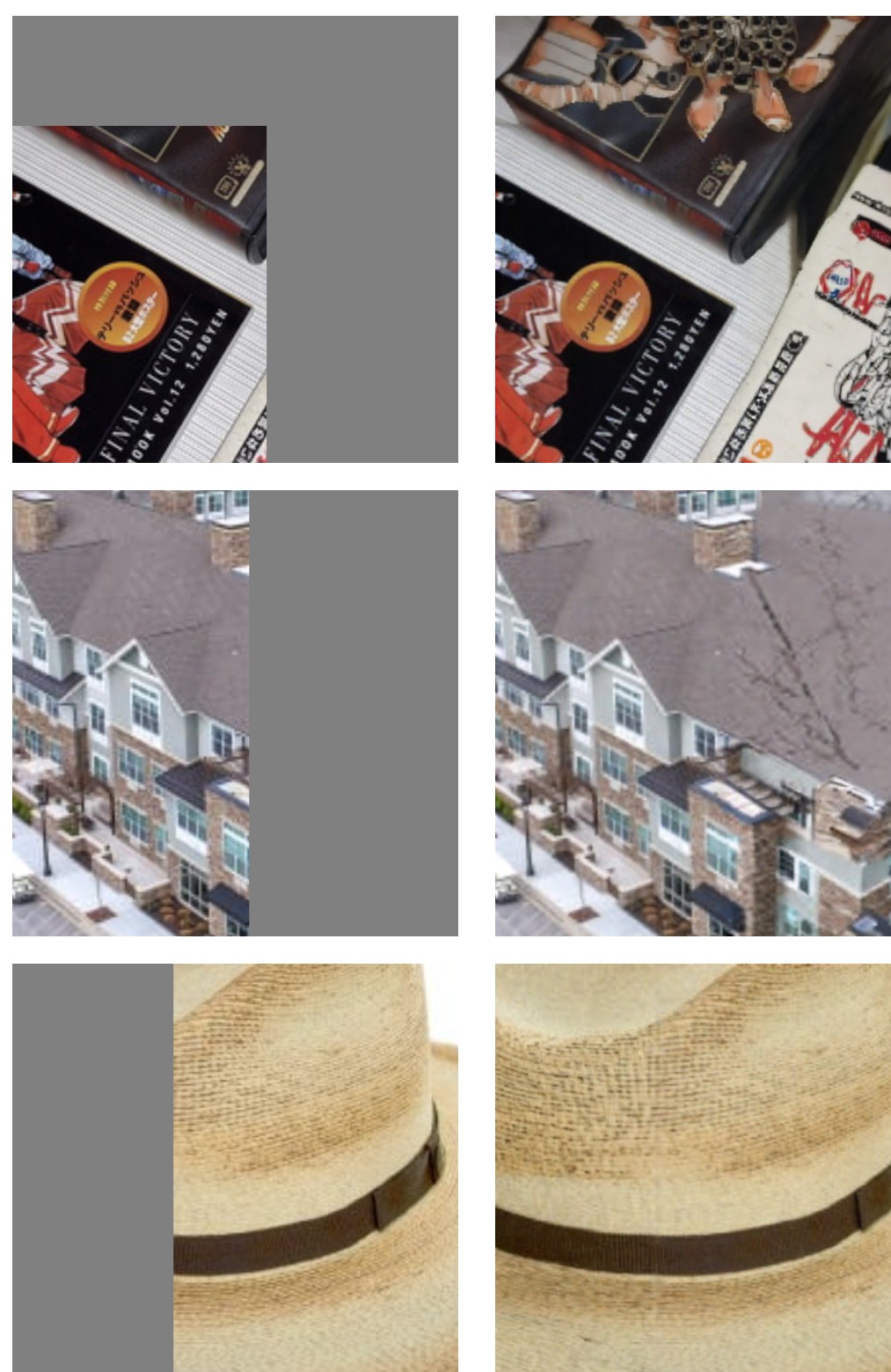

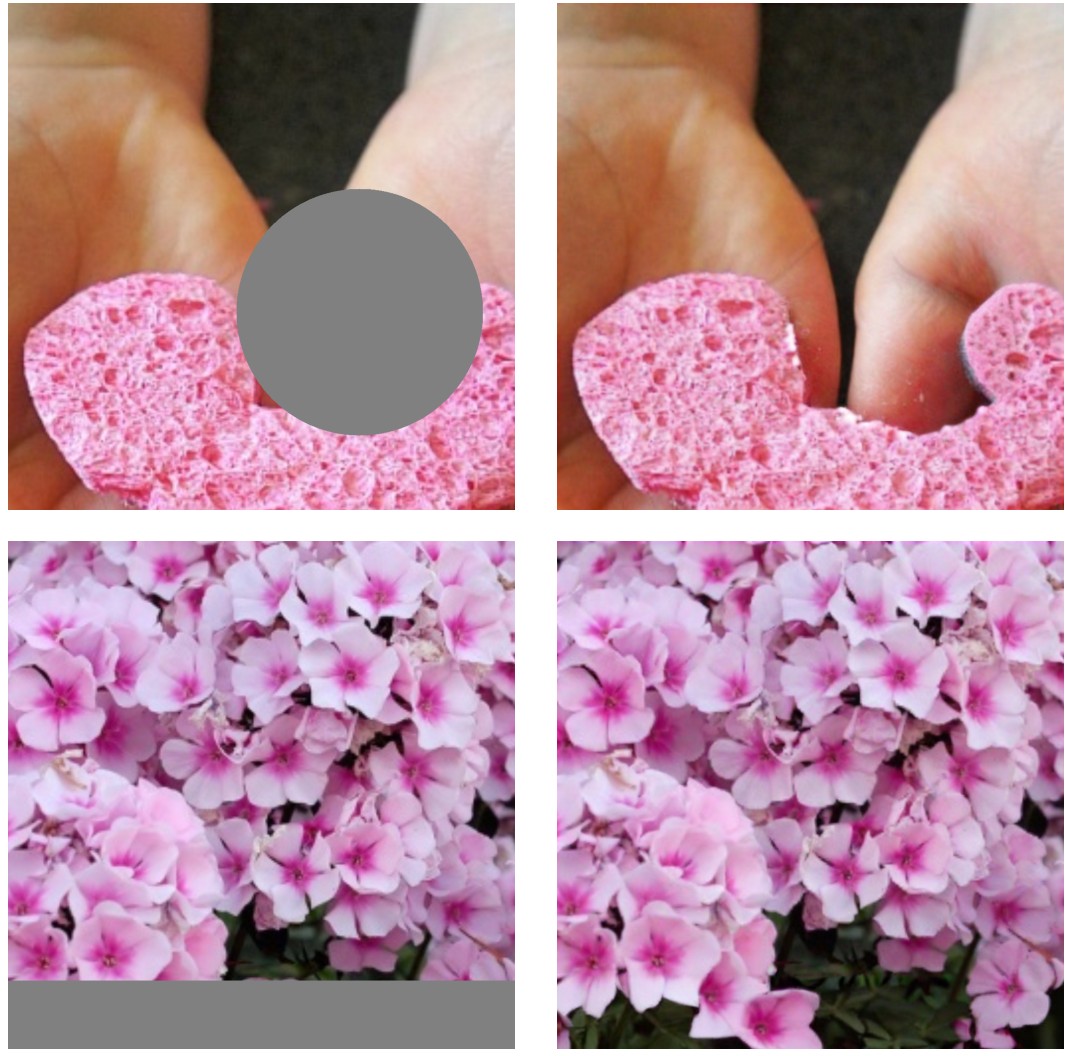

Figure 7: Inpainting examples. Left column: inputs by randomly masking images from the COYO dataset. Right column: inpainting outputs from our SDXL fine-tuned model with correlated noise.

with blurring corruption, leading to a blurred terminal distribution. Several other works have shown how to generalize diffusion models to find mappings between arbitrary input-output distributions, including [6, 10, 24]. One new finding in our work is that it is possible to start with a state-of-the-art model trained with white noise and fine-tune it easily to handle correlated noise. This allows us to convert vanilla diffusion models to Function Space Diffusion models by training them with noise sampled from Gaussian Processes. For more details, we refer the reader to Section 3.1.

# F    Experimental Details

## F.1    Dealing with Optical Flows

We use the RAFT model to predict the optical flows [83]. The optical flows can be computed with respect to the first frame or between subsequent frames. We find that the optical flow estimation is much better between subsequent frames and we use subsequent transformations to find the position in the original frame, whenever possible.

Since we are working with Latent Diffusion Models, all the warping happens in a lower-dimensional space. Fortunately, as observed in numerous prior works, including [57], there is a geometric

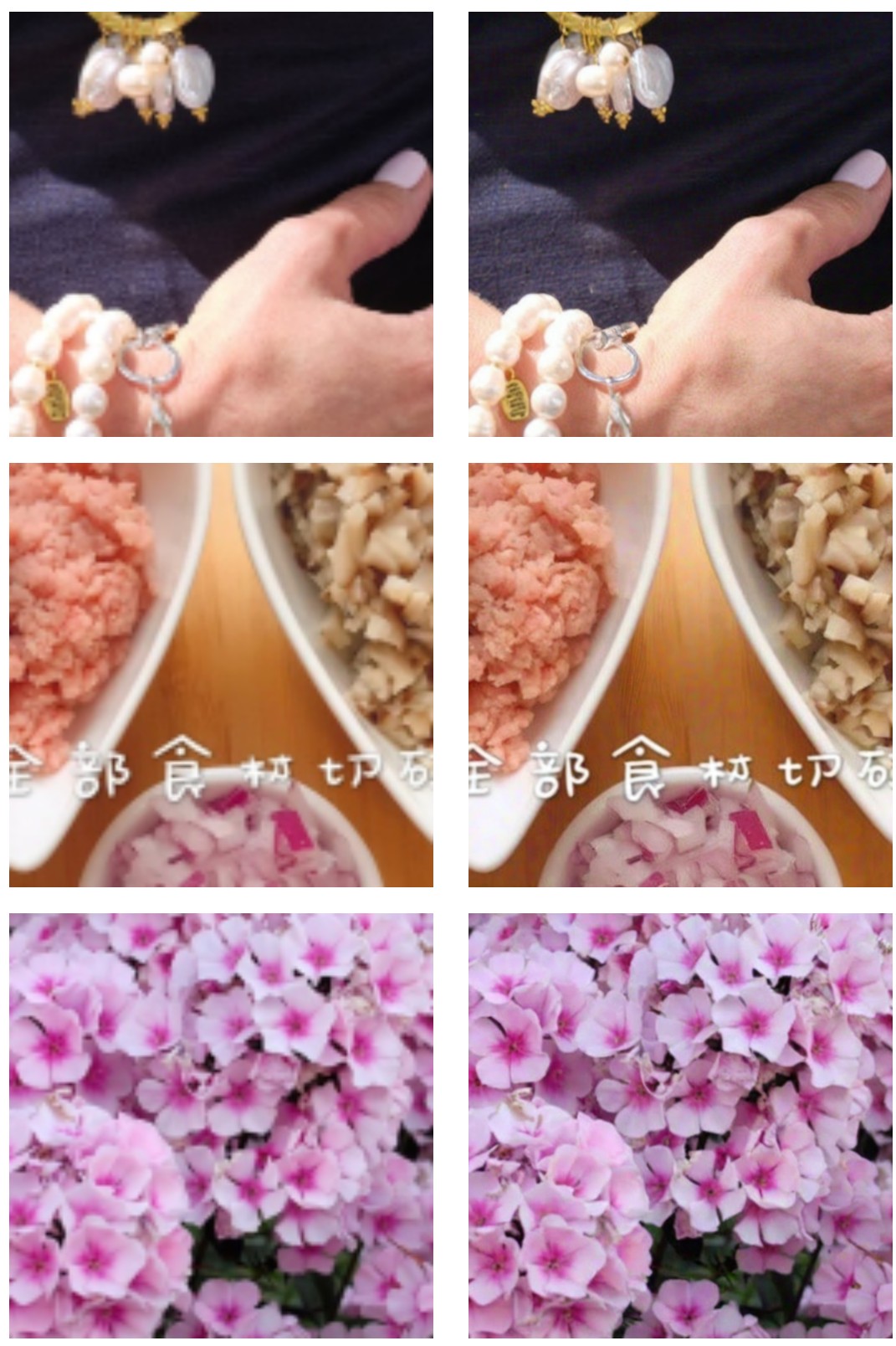

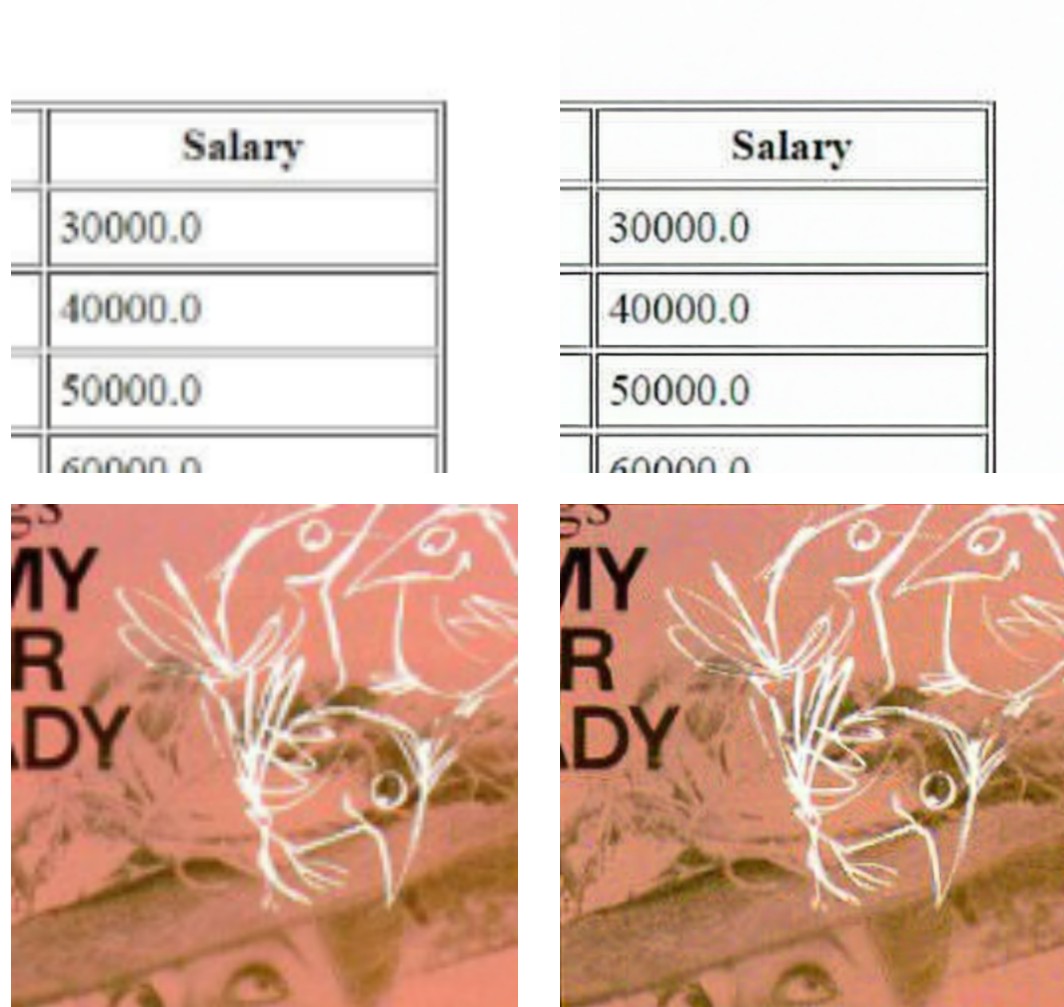

Figure 8: Super-resolution examples. Left column: downsampled inputs from the COYO dataset. Right column: super-resolution outputs from our SDXL fine-tuned model with correlated noise.

correspondence between pixel blocks and latent locations, i.e. pixel blocks are mapped to specific locations in latent space. This allows us to extract the flows from the input frames and convert them to optical flows for our latent vectors. Alternatively, one nat first map to latent space and then compute the optical flow there. We did not pursue this approach since we rely on a deep learning method for the flow-estimation and the underlying model has been trained on natural images.

## F.2 Stable Diffusion XL Finetuning

To fine-tune SDXL in conditional tasks, we use the reference implementation found in the following link: https://github.com/huggingface/diffusers/pull/6592. The reference implementation finetunes SDXL on the inpainting task, however, it is straightforward to adapt it to other conditional tasks, such as super-resolution. As mentioned in the paper, we train all our models for $100,000$ steps. We use the following training hyperparameters:

- Training resolution: $1024 \times 1024$.
- Batch size: $64$.
- Latent resolution: $128$.

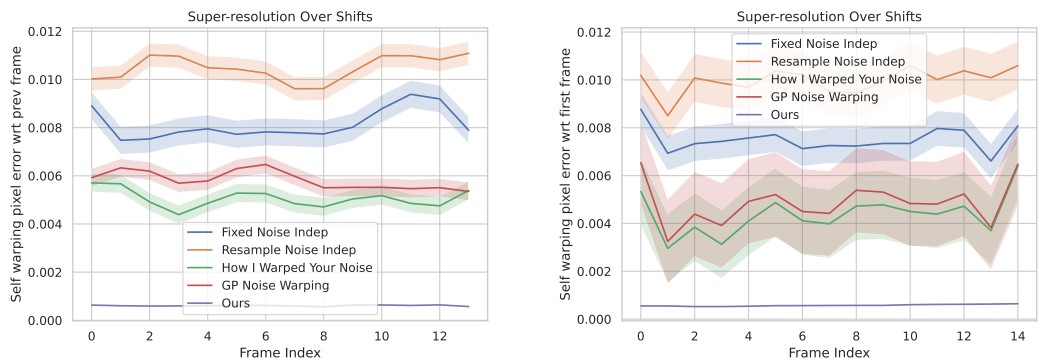

Figure 9: Schematic visualization of Equivariance Self Guidance (see Algorithm 1).

(a) Warping error w.r.t. previously generated frame in pixel space.

(b) Warping error w.r.t. first generated frame in pixel space.

Figure 10: Warping errors in pixel space for the super-resolution task as we shift the input frame.

- Optimizer Adam with Weight Decay. Optimizer parameters:
  - Learning rate: $5e - 6$
  - $\beta_1 = 0.9$
  - $\beta_2 = 0.999$
  - Weight Decay: $1e - 2$
  - $\epsilon = 1e - 08$
  - Max Gradient Norm (Gradient clipping): $1.0$
- Gaussian Process parameters:
  1. Truncation parameter: $2.0$
  2. Number of random features: $3000$
  3. Length scale: $0.004977$.

The parameter length scale controls the amount of correlation in the noise from the GP. Recall that RFFs are generated by sampling $z_j \sim N(0, \epsilon^{-2} I_2)$. To avoid aliasing artifacts when generating GP, we truncated the Normal distribution at $2\epsilon^{-1}$ (i.e., $2\times$ its standard deviation) and we made sure that $2\epsilon^{-1}$ is lower than the Nyquist–Shannon sampling frequency, i.e., $\frac{2\epsilon^{-1}}{2\pi} \leq \frac{\text{resolution}}{2}$. Given this, as

a general rule of thumb, we found that setting the length scale to be $\epsilon := \frac{2}{\pi \cdot \text{resolution}}$ leads to noise realizations that can be used to easily fine-tune Stable Diffusion XL.

We train all our models on 16 A100 GPUs on a SLURM-based cluster. The fine-tuning of the SDXL model on conditional tasks (super-resolution, inpainting) with correlated noise for 100k steps takes roughly 24 hours.

### F.3 Sampling Speed

Sampling guidance for equivariance increases the generation time for two reasons: i) we need to run more steps in order to make it effective and, ii) each step is more expensive since we need to perform an additional backpropagation. For our experiments, we use 50 steps instead of 25 steps that we use for unconditional sampling. Further, without guidance, we get $4.32$ iterations per second on a single A100 GPU while with guidance we obtain $1.62$ iterations per second.

The other hyperparameter used in sampling is the guidance strength, see Algorithm 1. For $\lambda = 0$, there is no guidance and the method just becomes GP Noise Warping. For higher $\lambda$ the gradient from the warping guidance becomes stronger. In our experiments, we found the value $\lambda = 1$ to perform the best. This is consistent with the choice of $\lambda$ in the Diffusion Posterior Sampling [15] paper which uses a guidance term to apply diffusion models for general inverse problems.

We perform all our sampling experiments on a single A-100 GPU. Without sampling guidance, it takes roughly 20 seconds to generate a single frame. We measure the performance of our method and the baselines on 2 second videos consisting of 16 frames.

## G   Broader Impact

Our method allows the use of image diffusion models to solve video inverse problems. There are both positive and negative societal implications of such a method. On the positive side, our method does not require training of video models which is typically expensive and contributes to increasing the AI carbon footprint. Further, democratizes access to video editing tools. The average practitioner can now leverage state-of-the-art image models to solve video inverse problems. To illustrate the effectiveness of our method, we trained powerful text-conditioned inpainting models that work on arbitrary images from the web. On the negative side, these models can be used for adversarial image and video editing. Further, our method can be used for the generation of deepfakes.

