# OpenReview forum: "Warped Diffusion: Solving Video Inverse Problems with Image Diffusion Models"
_NeurIPS.cc/2024/Conference — NeurIPS 2024 poster_

### Official Review · Reviewer_ebKQ · 2024-07-10

**Soundness:** 3
**Presentation:** 3
**Contribution:** 3
**Rating:** 6
**Confidence:** 4

**Summary:**

This paper aims to address video inverse problems using image diffusion models. By viewing videos as sequences of warping transformations between frames, the authors propose a novel approach called warped diffusion from a continuous function space perspective. Specifically, warped diffusion includes a Gaussian process-based noise-warping technique and a post-hoc equivariance self-guidance method to enhance temporal consistency. Experiments on video super-resolution and inpainting validate the effectiveness of warped diffusion. Moreover, warped diffusion first demonstrates superior performance with latent diffusion models such as SDXL.

**Strengths:**

1. The proposed method is well-motivated with well-constructed formulations.
2. The paper is well-written.
3. The visualization results are impressive.

**Weaknesses:**

1. As illustrated in Table 1 and Figure 3, the proposed GP Noise Warping performs worse than "How I Warped Your Noise" in terms of warping error. The core performance improvement appears to come from the equivariance self-guidance.
2. As mentioned in Section G.3, the equivariance self-guidance is inefficient. Although the proposed GP Noise Warping is more efficient than "How I Warped Your Noise," the overall process can be more time-consuming.

**Questions:**

1. "How I Warped Your Noise" operates in a zero-shot manner using pretrained video editing diffusion models without requiring additional training. Is it possible for warped diffusion to also function in a zero-shot manner?
2. I am curious about the effectiveness of warped diffusion with just the equivariance self-guidance alone, without the GP noise warping.
3. What would the performance be like if the equivariance self-guidance technique were integrated into "How I Warped Your Noise"? Would it surpass the current performance of warped diffusion?

**Limitations:**

The limitations and societal impacts have already been discussed in the paper.

---

> ### Author Rebuttal · Authors · 2024-08-06
>
> We thank the Reviewer for their time and their high-quality review. We are glad that the Reviewer appreciated the importance of the problem we are solving, the presentation of our work, and our experimental results. In what follows, we do our best to answer the questions raised by the Reviewer.
>
> `As illustrated in Table 1 and Figure 3, the proposed GP Noise Warping performs worse than "How I Warped Your Noise" in terms of warping error. The core performance improvement appears to come from the equivariance self-guidance.`
>
> We believe there is some misunderstanding here. In Table 1, no noise warping is happening (hence no comparison with the How I Warped Your Noise work). The reported results are for the generation of the first frame of a video. The results show that finetuning with correlated noise does not compromise performance. We will clarify this further to avoid confusing the reader.
>
> In Figure 3, we measure equivariance with respect to translation by an integer amount of pixels. For this deformation, the How I Warped Your Noise warping mechanism and the proposed GP Warping are essentially the same: both mechanisms just shift the input noise by the appropriate number of pixels. This is explained in Lines 315 to 318 in the paper. We intentionally chose to compare on this task because it simplifies the How I Warped Your Noise method's implementation (for which no reference code is available).
>
> Since for this task, GP Noise Warping and How I Warped Your Noise are essentially the same algorithm, the reason there is some small performance difference is that we are applying the warping to different models: the GP Noise Warping uses the finetuned model with correlated noise while the How I Warped Your Noise paper uses the finetuned model with independent noise. We will clarify this point in the paper and we thank a lot the Reviewer for raising it.
>
> `As mentioned in Section G.3, the equivariance self-guidance is inefficient. Although the proposed GP Noise Warping is more efficient than "How I Warped Your Noise," the overall process can be more time-consuming.`
>
> We agree with the Reviewer that the overall process can be more time-consuming if guidance is used. Our paper has two major contributions: (1) a principled framework for noise warping (GP Warping) and (2) a guidance method to enforce the equivariance of the underlying network (self-guidance). The How I Warped Your Noise paper is a warping method and hence when we compare the speed with this baseline, we measure the time needed for warping (see Lines 357). Our warping method (contribution (1)) is 16x faster and it generates images at higher resolution. That said, it is true that sampling guidance (contribution (2)) will significantly increase the sampling time. Namely, as we mention in Section G.3., we need ~2.67x the time needed without guidance for every single step and we run it for twice the number of steps, leading to an overall ~5.34x slowdown compared to not using guidance. For full transparency, we will move this discussion to the main paper and we thank the Reviewer for bringing up this important point.
>
> `"How I Warped Your Noise" operates in a zero-shot manner using pretrained video editing diffusion models without requiring additional training. Is it possible for warped diffusion to also function in a zero-shot manner?`
>
> This is a very important point and we thank the Reviewer for raising it. Our method cannot work in a zero-shot manner since it requires a model that is trained with correlated noise. We mention this in the paper, but we will explicitly add it to the Limitations Section (Section F). As we show in the paper, with minimal finetuning it is possible to get a model trained with independent noise to work with correlated noise (Table 1), but still, some amount of finetuning is required, as the Reviewer correctly pointed out.
>
> `I am curious about the effectiveness of warped diffusion with just the equivariance self-guidance alone, without the GP noise warping.`
>
> This is a great question. We ran some preliminary experiments for super-resolution on real-videos and we found that omitting the warping significantly deteriorates the results when the number of sampling steps is low. We found that increasing the number of sampling steps makes the effect of the initial noise warping less significant, at the cost of increased sampling time. We will include this discussion in the camera-ready version of our paper and we thank a lot the Reviewer for raising this important point.
>
> `What would the performance be like if the equivariance self-guidance technique were integrated into "How I Warped Your Noise"? Would it surpass the current performance of warped diffusion?`
>
> This is an excellent question. We believe that adding self-guidance to the How I Warped Your Noise paper would significantly improve the results. Unfortunately, there is no way to check this for general transformations because there is no reference implementation for us to try. We tried this for the integer shift transformation (for which the implementation becomes trivial) and the How I Warped Your Noise + self-guidance method matched the performance of our proposed algorithm. We will include this experiment in the camera-ready version of our work and we thank a lot the Reviewer for raising this insightful question.
>
> We hope our response addresses your concerns and that you consider increasing your rating.

---

> ### Comment · Reviewer_ebKQ · 2024-08-10
>
> I appreciate the authors' efforts in addressing my concerns. After reviewing the other reviewers' comments and the authors' responses, I have decided to maintain my original score.
>
> I would like to see the authors include the discussions and experiments in the response to their final revision.

---

> > ### Author Response · Authors · 2024-08-10
> > **Thank you for acknowledging our rebuttal**
> >
> > We thank the Reviewer for acknowledging our rebuttal. We will make sure to include the discussions and experiments in the camera-ready version of our work.
> >
> > We thank the Reviewer once again for their time and for helping us improve our work!

---

### Official Review · Reviewer_kW4a · 2024-07-13

**Soundness:** 3
**Presentation:** 3
**Contribution:** 3
**Rating:** 6
**Confidence:** 3

**Summary:**

This paper addresses the challenge of temporally correlated inverse problems by employing an image diffusion model. The authors introduce a technique termed Warped Diffusion, which incorporates an equivariance self-guidance mechanism. This mechanism ensures that the generated frames maintain consistency when subjected to warping transformations. The paper demonstrates the application of this technique in the context of video inpainting and video super-resolution through a series of experiments.

**Strengths:**

1. The paper is presented in a structured manner, with illustrations that aid in understanding. The content is coherent and follows a logical sequence.

2. The authors offer an analysis of noise warping and equivariance self-guidance, providing both intuitive explanations and theoretical underpinnings.

3. 3. The experiments are extensive to support the claim.

**Weaknesses:**

1. The authors report the noise warping speed at Line 357. We recommend that the authors also include the complete inference time required to process an entire video, which would provide a more comprehensive understanding of the method's efficiency.

2. We suggest that the authors provide more video results to further demonstrate the capabilities and limitations of the proposed method in various scenarios.

3. It is observed that the inpainted regions in the video inpainting examples remain static, such as the cat in the video not exhibiting motion. All apparent motion seems to be attributed to camera movements, with no optical flow present within the inpainted regions. We would appreciate an explanation for this phenomenon and its implications on the video inpainting process.

**Questions:**

See the weaknesses.

**Limitations:**

The authors have adequately discussed the limitations and negative societal impact in the paper.

---

> ### Author Rebuttal · Authors · 2024-08-06
>
> We thank the Reviewer for their time and insightful review. We are very glad that the Reviewer appreciated the presentation of our work, the theoretical underpinnings of our formulation and the strong empirical performance of our method.
>
> `The authors report the noise warping speed at Line 357. We recommend that the authors also include the complete inference time required to process an entire video, which would provide a more comprehensive understanding of the method's efficiency.`
>
> We have some additional discussion regarding sampling time in Section G.3. of the Appendix. Processing a 2-second video takes roughly ~5 minutes on a single A-100 GPU. We will move this discussion to the main paper, as recommended by the Reviewer.
>
> `We suggest that the authors provide more video results to further demonstrate the capabilities and limitations of the proposed method in various scenarios.`
>
> We thank the Reviewer for their suggestion. We include more results on the [project webpage](https://anonneurips2024.github.io/) as recommended by the Reviewer. We will de-anonymize this webpage and make it available to the public upon acceptance of our work.
>
> `It is observed that the inpainted regions in the video inpainting examples remain static, such as the cat in the video not exhibiting motion. All apparent motion seems to be attributed to camera movements, with no optical flow present within the inpainted regions. We would appreciate an explanation for this phenomenon and its implications on the video inpainting process.`
>
> We thank the Reviewer for raising this question. Our proposed framework is not constrained to static examples and can work with real optical flows, as demonstrated for the super-resolution task. Inpainting is a much harder inverse problem that super-resolution and thus maintaining consistency for real flows might be more challenging. We did not experiment with real video inpainting in this paper but we plan to include some examples for the camera-ready version.
>
> We hope our response addresses your concerns and that you consider increasing your rating.

---

### Official Review · Reviewer_WsM3 · 2024-07-24

**Soundness:** 3
**Presentation:** 2
**Contribution:** 2
**Rating:** 5
**Confidence:** 4

**Summary:**

The paper follows a previous work on noise-warping methods for video inverse problem. The authors proposed instead of giving temporally consistent noise maps to the DM, using a continuous function space representation for DM directly can serve more complex spatial transformations and results in better temporal consistency in the generated videos. They demonstrated the proposed method on video inpainting and super-resolution.

**Strengths:**

The proposed method is novel in terms of using Gaussian processes to handle the noise function so that no mapping is required for the noise. The authors derived the equivariance for deformation and by satisfying this the gaussian process is used to guide the generation to be consistent under the warping transformation. To avoid additional training, a sampling mechanism was proposed to ensure this satisfaction at inference time.

**Weaknesses:**

The key statement in the paper is the importance of the equivariance of the DM to achieve temporal consistancy. The authors argued that the previous work "How I Warped Your Noise" performed badly when the DM is not equivariant. However, there are no explicit examples or solid proof on this statement. The proposed method for noise warping is more like offering another perspective than showing the failing reasons of the other work.

Furthermore, for super-resolution experiments, the authors did not provide comparison against "How I warped your noise". Even though they elaborated that the authors of "How I warped your noise" acknowledged to them that the work does not result in temporal consistency, I believe it is still better to include the comparison given how many super-resolution cases were presented in that work. If they don't provide the code, the authors may consider modify the one they have for the inpainting experiments and list out the comparison for more convincing results.

**Questions:**

1. It is not clear to me why the function space diffusion models need to be equivariant with respect to the underlying spatial transformations. I can imagine the necessity, but in your derivation, I don't see the proof that you stated you provide. Can you provide explicit examples or solid proof on this statement?
2. Why in fig 3 how i warped your noise seems to be in the second place in terms of error, but in table 2, the method seems not to outperform any other baselines?

**Limitations:**

The authors did not address any limitations

---

> ### Author Rebuttal · Authors · 2024-08-06
>
> We thank the Reviewer for their time and their valuable review. We are glad that the Reviewer appreciated the novelty of our work. In what follows, we do our best to answer some remaining questions.
>
> `The authors argued that the previous work "How I Warped Your Noise" performed badly when the DM is not equivariant. However, there are no explicit examples or solid proof on this statement. The proposed method for noise warping is more like offering another perspective than showing the failing reasons of the other work.`
>
> Our framework has two components: (1) the GP warping scheme, and (2) the self-guidance method at inference time. In Figure 3 of the paper, we show that (1) is not enough: both GP noise warping and the How I Warped Your Noise baseline still have significant warping error. Self-guidance is essential to further decrease the warping error because the DM is not equivariant. This is also illustrated in Row 2 of Figure 1: the How I Warped Your Noise warping scheme fails to produce consistent results because the underlying DM is not equivariant. If the Reviewer wants further clarification on this point, we would be happy to provide it.
>
> `Furthermore, for super-resolution experiments, the authors did not provide comparison against "How I warped your noise". Even though they elaborated that the authors of "How I warped your noise" acknowledged to them that the work does not result in temporal consistency, I believe it is still better to include the comparison given how many super-resolution cases were presented in that work. If they don't provide the code, the authors may consider modify the one they have for the inpainting experiments and list out the comparison for more convincing results.`
>
> We thank the Reviewer for their suggestion! We provide comparisons for the super-resolution task with How I Warped Your Noise in the case of integer translation in the following anonymous links:
> 1) [(Self-) Warping Error with respect to previous generated frame in pixel space](https://anonneurips2024.github.io/assets/self_warping_pixel_error_wrt_prev_frame.pdf)
> 2) [(Self-) Warping Error with respect to first frame in pixel space](https://anonneurips2024.github.io/assets/self_warping_pixel_error_wrt_first_frame.pdf)
>
> These results are similar to what's given in Figure 3 of the paper and will be included in the camera-ready version of our paper. Once again, we thank the Reviewer for suggesting this important experiment.
>
> `Why in fig 3 how i warped your noise seems to be in the second place in terms of error, but in table 2, the method seems not to outperform any other baselines?`
>
> We believe the Reviewer might have misunderstood this part. Figure 3 shows the Warping Error and the How I Warped Your Noise paper gets the second (best) place. This is consistent with the first column (Warping Error) of Table 2 which places again the GP in the second best place in terms of warping error. The rest of the columns of Table 2 measure the quality of the generated images (but not temporal consistency across frames).
> Any warping mechanism sacrifices some quality for temporal consistency. This has been also reported in the How I Warped Your Noise paper (see Table 1, page 7 of the paper). Our results in Table 2 are consistent with this finding. Hopefully, this clarifies the question raised by the Reviewer.
>
> `The authors did not address any limitations`
>
> We would like to bring to the Reviewer’s attention Section F of our Appendix. In this Section, we state several limitations of our work. To increase the visibility of this Section, we will bring it to the main paper in the next-revision of our work.
>
> `It is not clear to me why the function space diffusion models need to be equivariant with respect to the underlying spatial transformations.`
>
> The formalization of this statement follows from the definition of the video that we use in this paper. We define a video such that the next frame is given as a deformation of the last frame. If the generative model is not equivariant w.r.t. this transformation, then its output will no longer satisfy this definition of a video. We argue that since optical flows are a useful tool for video modeling and have been empirically successful, this definition of a video, through the optical flow, is a reasonable one. In terms of empirical proof, Figure 1 shows that the robot keeps changing if you don't have equivariance.
>
> **We hope our response addresses your concerns and that you consider increasing your rating.**

---

> > ### Comment · Reviewer_WsM3 · 2024-08-13
> >
> > Thank you for your response, especially for providing the comparisons for the super-resolution task and clarifying my misunderstanding on table 2. However, I am still not convinced about the "equivariant" statement. I understand your formulation and approach, but I don't agree other approaches "failure" automatically prove your statement.

---

> > > ### Author Response · Authors · 2024-08-14
> > > **Further Clarifications**
> > >
> > > We agree with the reviewer that we have not proven other approaches will not work. Indeed, let us consider the following argument. We'll work with $H$ a Hilbert space of real-valued functions over the domain $D = [0,1]^2$. Consider first the following definition of a video: A two-frame video is a pair of functions $f_0, f_1 \in H$ such that there exists a bounded, injective map $T: D \to \mathbb{R}$ with the property that $f_1(x) = f_0 \big ( T^{-1}(x) \big )$ for all $x \in T(D) \cap D$.
> > >
> > > Now let $G: H \to H$ be a generative model for which there exists some $\xi_0 \in H$ such that $G(\xi_0) = f_0$. Then following holds: if $G$ is equivariant w.r.t. to $T$ then the pair $\big ( f_0, G(\xi_0 \circ T^{-1}) \big )$ is a two-frame video. This follows from the definition of equivariance as $G(\xi_0 \circ T^{-1}) = G(\xi) \circ T^{-1} = f_0 \circ T^{-1} $ on $T(D) \cap D$. Certainly, the other direction does not hold true. In particular, it does not follow that if $\big ( f_0, G(\xi_0 \circ T^{-1}) \big )$ is a two-frame video then $G$ is equivariant w.r.t. $T$. Therefore we agree with the reviewer that we have not proven that equivariance is necessary, however, we have proven that it is sufficient. We will explicitly point this out in the camera-ready version.
> > >
> > > This lack of necessity stems from the fact that our definition of a video is weak and allows many possible pairs to be videos. One natural way of strengthening it is as follows: Let $\mathcal{T}$ denote the set of bounded, injective maps on $D$ then a two-frame video is a pair of functions $f_0,f_1 \in H$ such that the following problem admits a unique maximizer
> > > \[\max_{T \in \mathcal{T}} \big \{ |T(D) \cap D| : f_1(x) = f_0 \big ( T^{-1}(x) \big ) \text{ for all } x \in T(D) \cap D \big \}.\]
> > > In particular, we enforce that there is a unique deformation which keeps the maximum number of pixels in frame. From uniqueness, it now follows that: the pair $\big ( f_0, G(\xi_0 \circ T^{-1}) \big )$ is a two-frame video if and only if $G$ is equivariant w.r.t. to $T$.
> > >
> > > We thank the reviewer for bringing this to our attention and are happy to discuss the mathematical modeling of videos in the camera-ready version further.
> > >
> > > If that's not what the Reviewer was looking for, please let us know and we will try our best to incorporate the Reviewer's feedback in our camera-ready.

---

### Official Review · Reviewer_zjYi · 2024-07-30

**Soundness:** 3
**Presentation:** 2
**Contribution:** 3
**Rating:** 5
**Confidence:** 3

**Summary:**

This paper proposes a self-guidance equivariance approach using the Image Diffusion model for generating temporally consistent videos. In previous work HOW I WARPED YOUR NOISE[1], noise is warped across frames to ensure the noise maps are temporally consistent. However, temporally consistent noise maps do not guarantee temporally consistent output. In this work, the authors address this issue. The proposed self-guidance equivariance enforces that the generated frames remain consistent under the warping transformation. The proposed method can generate more consistent results. In their experiments, the authors conducted comparisons on video inpainting and superresolution. It is worth noting that the proposed method outputs consistent content in the inpainting area across frames.

**Strengths:**

- Problem Setup
  - The problem seems reasonable and important, especially in the case of video inpainting. Compared to other video applications, any temporary inconsistent case can create a dramatic failure case. See Figure 1.

- Temporal consistent results.
    - In this work, the authors show how the proposed equivariance self-guidance improves video inpainting results and superresolution.

- More efficient noise warping.
   - Compared to the previous work [How I Warped Your Noise], this work improves the inference time by up to 16 times.

**Weaknesses:**

- Better illustration and explanation of how the `self-guidance Equivariance` works.
  The main novelty of the paper is the self-guidance equivariance, which improves performance compared to previous work, HOW I WARPED YOUR NOISE [1]. However, it is not easy to understand how it works in Figure 2. The current Figure 2 only shows its role and the input and output in the overall system but does not describe its function. Besides, this figure seems to provide more information about how to use optical flow to guide the warping process of noise.

- Inpainting examples and usefulness?
  Do the warped noise and self-guidance Equivariance work for generating "moving objects" in video inpainting? According to the formulation and the generation results, it seems it will favor generating "still objects" in the video inpainting case because it will easily meet the "warped" mechanism.

- Experiment Design.
  Experiments in Section 4.1 are not needed (or do not provide a useful argument) in the paper. Previous works have shown that warped noise is better than independent noise across the frames in a video. The only contribution here is that the fine-tuned version performs better than the un-fine-tuned version. The finding seems unrelated to the paper or provides quite limited information.

**Questions:**

- A better way to illustrate how the proposed Equivariance Self-Guidance works.
  - A proper figure would be better instead of just formulations.

- See Point 2 in Weakness; can the proposed method handle cases where the object is moving or has different actions in the video inpainting case instead of being still?

**Limitations:**

For the limitations, the authors mention that in some situations, the work may fail. It would be helpful to also provide the corresponding failing examples, such as the second and third examples mentioned in Appendix F Limitations.

---

> ### Author Rebuttal · Authors · 2024-08-06
>
> We thank the Reviewer for their thoughtful review. We are glad to see that the Reviewer appreciated many aspects of our submission, such as the importance of the problem we are addressing and the experimental benefits we are obtaining. In what follows, we do our best to address any remaining concerns.
>
> `[...] it is not easy to understand how it works in Figure 2. [...] A proper figure would be better instead of just formulations.`
>
> We thank the Reviewer for their comment. We agree that Figure 2 should be improved. Following the recommendation of the Reviewer, we updated the figure to show that the self-guidance is applied in each step of the ODE sampler. The figure is now available at the (anonymous) project website and at [this link](https://bashify.io/i/veUk9P). We plan to include this update in the camera-ready version of our work. If there is any more feedback, we would be glad to incorporate it.
>
> `Do the warped noise and self-guidance Equivariance work for generating "moving objects" in video inpainting? According to the formulation and the generation results, it seems it will favor generating "still objects" in the video inpainting case because it will easily meet the "warped" mechanism.`
>
> We thank the Reviewer for their insightful question.  The proposed framework should work for moving objects in video inpainting as long as the optical flow map is available and there are no occlusions. That said, as the Reviewer correctly points out, for realistic videos, the estimation of the optical flow map might be noisy. We will make this point more explicit in our paper and we will include videos of moving objects in the camera-ready version of our work. We thank again the Reviewer for bringing up this important point.
>
> `Experiments in Section 4.1 are not needed (or do not provide a useful argument) in the paper. Previous works have shown that warped noise is better than independent noise across the frames in a video. The only contribution here is that the fine-tuned version performs better than the un-fine-tuned version. The finding seems unrelated to the paper or provides quite limited information.`
>
> We respectfully disagree with the Reviewer on this point. Typically, diffusion models are trained with i.i.d. noise. In Section 4.1, we show that one can finetune a state-of-the-art diffusion model to work with correlated noise as input. Before finetuning, Stable Diffusion XL will produce unrealistic images when the sampling chain is initialized with correlated noise. Our experiments show that post-finetuning, the model can handle spatially correlated noise in the input without compromising performance. To the best of our knowledge, spatially correlated noise has not been shown effective in prior work (note that this is different from prior works on video generation that use temporally correlated noise). Our GP Warping mechanism requires models that can handle correlated noise. Hence, these fine-tunings are essential for the rest of the paper. We will clarify this point in the main paper to avoid confusing the reader.
>
> `For the limitations, the authors mention that in some situations, the work may fail. It would be helpful to also provide the corresponding failing examples, such as the second and third examples mentioned in Appendix F Limitations.`
>
> We definitely agree with the Reviewer and we thank them for encouraging us to make negative results publicly available.
>
> * Equivariance of the decoder. Sometimes, the decoder is not equivariant with respect to the underlying transformation. We noticed that this is a common failure for text rendering. This can be better illustrated with an example. The generated latent video [here](https://anonneurips2024.github.io/assets/videos/3_output_latent_video.mp4) appears equivariant. However, the decoded version of this video, found [here](https://anonneurips2024.github.io/assets/videos/3_output_video.mp4) is not equivariant. Specifically, the produced text changes from one frame to the other.
>
> * Correlation artifacts: For extreme deformations, there is a distribution shift between training and inference which leads to correlation artifacts. This has already been observed in the prior work, How I Warped Your Noise. In [this](https://warpyournoise.github.io/docs/assets/videos/SuperRes/Bear/adv.mp4) example from the project website, there appears to be a colored wave near the back of the bear. This is an example of a correlation artifact. Such artifacts also appear sometimes in our work, e.g. see the texture artifacts that appear in the final frame of [this](https://anonneurips2024.github.io/assets/videos/216_output_video.mp4) generated video.
>
> We will definitely include these failure cases in our camera-ready version and we thank again the Reviewer for raising this important point. **We hope our response addresses your concerns and that you consider upgrading your rating.**

---

> > ### Comment · Reviewer_zjYi · 2024-08-14
> >
> > Thanks for the clarification, and I'm happy to see the improved version. I might be confused by the wording in Section 4.1. I understand that performance improves after finetuning, which is already useful across all machine learning tasks and areas. This improvement is expected. Given the limited space, I would prefer combining Section 4.1 with other ablation studies and including more diverse experiments in the full paper.

---

> > > ### Author Response · Authors · 2024-08-14
> > > **Thank you for acknowledging our rebuttal**
> > >
> > > Dear Reviewer,
> > > thank you for reading our rebuttal; we are happy you appreciated the improved version. If there are no other major concerns, we would greatly appreciate it if you could increase your rating of our submission.
> > >
> > > We understand your suggestion now and will incorporate it in the camera-ready version of our work to improve the presentation of our work.
> > >
> > >
> > > Thank you again for your time and for helping us to strengthen our work!

---

### Decision · Program_Chairs · 2024-09-25

**Decision:**

Accept (poster)

**Comment:**

All the 4 reviewers recommend acceptance. The authors should add discussions on failure cases, improve the organization and the presentation of the manuscript, and include the additional results and clarifications into the final copy.